# Epididymal epithelium propels early sexual transmission of Zika virus in the absence of interferon signaling

Alexander G. Pletnev[1], Olga A. Maximova[1], Guangping Liu[1], Heather Kenney[1], Bianca M. Nagata[2], Tatiana Zagorodnyaya[3], Ian Moore[2], Konstantin Chumakov [3] & Konstantin A. Tsetsarkin [1✉]

Recognition of Zika virus (ZIKV) sexual transmission (ST) among humans challenges our understanding of the maintenance of mosquito-borne viruses in nature. Here we dissected the relative contributions of the components of male reproductive system (MRS) during early male-to-female ZIKV transmission by utilizing mice with altered antiviral responses, in which ZIKV is provided an equal opportunity to be seeded in the MRS tissues. Using microRNA-targeted ZIKV clones engineered to abolish viral infectivity to different parts of the MRS or a library of ZIKV genomes with unique molecular identifiers, we pinpoint epithelial cells of the epididymis (rather than cells of the testis, vas deferens, prostate, or seminal vesicles) as a most likely source of the sexually transmitted ZIKV genomes during the early (most productive) phase of ZIKV shedding into the semen. Incorporation of this mechanistic knowledge into the development of a live-attenuated ZIKV vaccine restricts its ST potential.

[1] Laboratory of Infectious Diseases, National Institute of Allergy and Infectious Diseases, National Institutes of Health, Bethesda, MD, USA. [2] Infectious Disease and Pathogenesis Section, Comparative Medicine Branch, National Institute of Allergy and Infectious Diseases, National Institutes of Health, Rockville, MD, USA. [3] Center for Biologics Evaluation and Research, U.S. Food and Drug Administration, Silver Spring, MD, USA. ✉email: tsetsarkinka@niaid.nih.gov

The increasingly recognized ability of some arboviruses to utilize a non-vector-borne human-to-human mode of transmission via sexual contact[1–6] presents new challenges to the understanding of virus maintenance in nature. To get ahead in developing preventive and mitigating measures to control viral outbreaks, extensive research to better understand such unusual viral tropism is needed. Zika virus (ZIKV) is a mosquito-borne flavivirus that recently emerged in outbreaks on the Pacific islands followed by a large-scale epidemic in the Central and South America (reviewed in ref. [7]). Typically transmitted by infected *Aedes* mosquitoes, ZIKV has also been shown to disseminate among humans via male-to-female sexual contact[1,8–10] (reviewed in refs. [5,11,12]). The majority of infected men (>50%) shed ZIKV RNA in semen. The shedding peaks between 8 and 30 days after onset of the illness, followed by gradual viral clearance, which in some individuals takes up to 10 months[13–15]. Infectious ZIKV has also been recovered from a number of semen samples, although seminal clearance of the virus occurs faster than that of the viral RNA[13,15,16].

Despite substantial research efforts, a clear understanding of the cellular and molecular mechanisms underlying ZIKV male-to-female ST is yet to emerge. A number of cell types within different parts of the male reproductive system (MRS) can support ZIKV replication, although susceptibility of these cells to ZIKV infection varies significantly among different MRS organs. In mice, ZIKV replication in testis and epididymis begins with viral infection of cells located in the interstitium of these organs, followed by robust replication in cells that constitute seminiferous tubules of the testis and in epididymal epithelium[17–24]. In the prostate, ZIKV targets the organ's epithelium cells, although to a lesser extent than in the epithelium of the epididymis[23,25]. In mouse seminal vesicles, ZIKV replicates to a considerably lower titer than in the testis and epididymis[20,21,26], however, specific cellular targets of ZIKV in this organ have not been characterized. ZIKV RNA and/or viral antigen have also been detected in many parts of the MRS of new-world and old-world nonhuman primates (NHPs)[25,27–30]. Studies in NHPs tend to indicate a higher load and longer persistency of ZIKV RNA in the testis and epididymis, as compared to other MRS organs[28,30]. This incriminates testis and/or epididymis as a possible source of sexually transmitted ZIKV, which is consistent with reports of a prolonged detection of ZIKV antigen in the sperm and non-sperm cells in the human semen[31–33]. However, a high variability of the results obtained in NHPs makes it challenging to identify the precise role that each individual MRS component plays in ZIKV infection. Moreover, ZIKV was also detected in the semen of vasectomized men[34–36] and mice[19], suggesting that infection of the accessory glandular organs of the MRS is sufficient to support ZIKV replication and its seminal shedding independently of testicular and epididymal infections.

Here, we aimed to elucidate the relative significance of different cell types within the MRS of mice with altered antiviral responses as a source of ZIKV virions transmitted to the female sexual partner. For that, we used two independent yet complementary strategies that allowed us (i) to block ZIKV replication in various parts of the MRS by microRNA(miRNA)-targeting of viral genome and (ii) to trace the MRS tissue-dependent ZIKV progenies by the unique molecular identifiers (barcodes) incorporated into viral genome.

## Results

**Organ-specific restriction of ZIKV replication in the MRS by miRNA targeting.** Presumably, all components of the MRS may be seeded by ZIKV from the blood, support virus replication, and contribute to the viral shedding into the final product of the MRS—the ejaculate. ZIKV replication can be selectively blocked in a tissue of interest by the modification of the viral genome with sequences complementary to a miRNA specific for (or highly enriched in) that tissue (i.e., miRNA-targeting)[22]. It was shown that the mir-202-5p (hereafter mir-202) is expressed in the seminiferous tubules of the testis (Sertoli and spermatogenic cells), but not in any other MRS tissue components (Fig. 1a)[22,37–39]. The precise role of mir-202 in testicular physiology has yet to be described. Conversely, the mir-141-3p (hereafter mir-141) is widely expressed in the epididymis (epithelial cells), accessory glands (SV/P) (Fig. 1b), and the epithelium of the MRS ductal components such as vas deferens (Fig. S1), but not in the testis (Fig. 1b)[22,40–42]. Studies showed that mir-141 regulates maintenance of the epithelial cell phenotype by controlling the expression of E-cadherin[43,44]. This implies that mir-141 might be similarly expressed in all tissue components of the MRS that are comprised of the epithelial cells (i.e., epididymis, accessory glands, and ducts).

Previously, using 4–6 weeks old AG129 male mice that lack type I and type II interferon receptor (IFNR) genes, we showed that ZIKV genome targeting for mir-202 effectively restricts ZIKV replication in the cells located inside of seminiferous tubules, which blocks ZIKV migration from testis to epididymis[22]. Similarly, mir-141 targets restrict ZIKV infection of epididymal epithelium. Since most mouse strains reach sexual maturity after 6 weeks of age, in the present study, we used AG129 male mice of 10–20 weeks of age (hereafter, adult male mice). These mice were infected intraperitoneally (IP) with $10^6$ plaque forming units (pfu) of the mir-202-targeted [designated 2 × 202(T)] or mir-141-targeted [designated 2 × 141(T)] ZIKV constructs (Fig. S2), followed by analysis of viral replication in the MRS. We elected to dissect seminal vesicle and prostate as one specimen (designated as seminal vesicle/prostate or SV/P) since a precise anatomical separation of seminal vesicle from anterior prostate did not seem feasible and would be compromised by viral cross-contamination (Fig. S3). As a control, we infected mice with 2 × scr virus (contains scramble (scr) sequences at the sites of miRNA-targets), or with a parental ZIKV-NS3m virus (Fig. S2)[22,45]. Since miRNA-targeted viruses can accumulate escape mutants in the organs expressing a given miRNA, the regions containing scr sequences or miRNA targets were sequenced in all tissue-isolated viruses (with virus titers >10-fold over the limit of detection). If a deletion or a point mutation was detected in the region of miRNA target insertion, the titers for the stable (stb) or mutated (mut) viruses were presented separately.

Previously, we showed that in AG129 mice, the viremia with 2 × scr or ZIKV-NS3m virus peaks at 1 day post infection (dpi), and subsides rapidly thereafter[22]. At 1 dpi, mice inoculated with miRNA-targeted viruses developed a high level of viremia, which was comparable to that of 2 × scr-infected or ZIKV-NS3m-infected mice (Fig. 1c; $p > 0.05$, one-way ANOVA), suggesting that all viruses would have an equal opportunity to seed mouse tissues via a hematogenous route.

Seeding of ZIKV in the testis and epididymis begins with infection of the cells located in the interstitium of both organs[17,22,24]. Subsequently, infection progresses to the cells located inside the seminiferous tubules (expressing the mir-202) or epididymal epithelium (expressing the mir-141). It was shown that in the epididymis, progression of ZIKV infection from the interstitium to epithelium occurs ~2–4 days earlier than progression of infection to seminiferous tubules in the testis[21,46–48]. Replication of 2 × 202(T) and 2 × 141(T) viruses in the testis and epididymis (Fig. 1d, e) was consistent with the expression of mir-202 and mir-141 in these organs (Fig. 1a, b). Between 9 and 17 dpi, accumulation of 2 × 202(T)stb [but not 2 × 202(T)mut or 2 × 141(T)] was strongly inhibited in the testis as

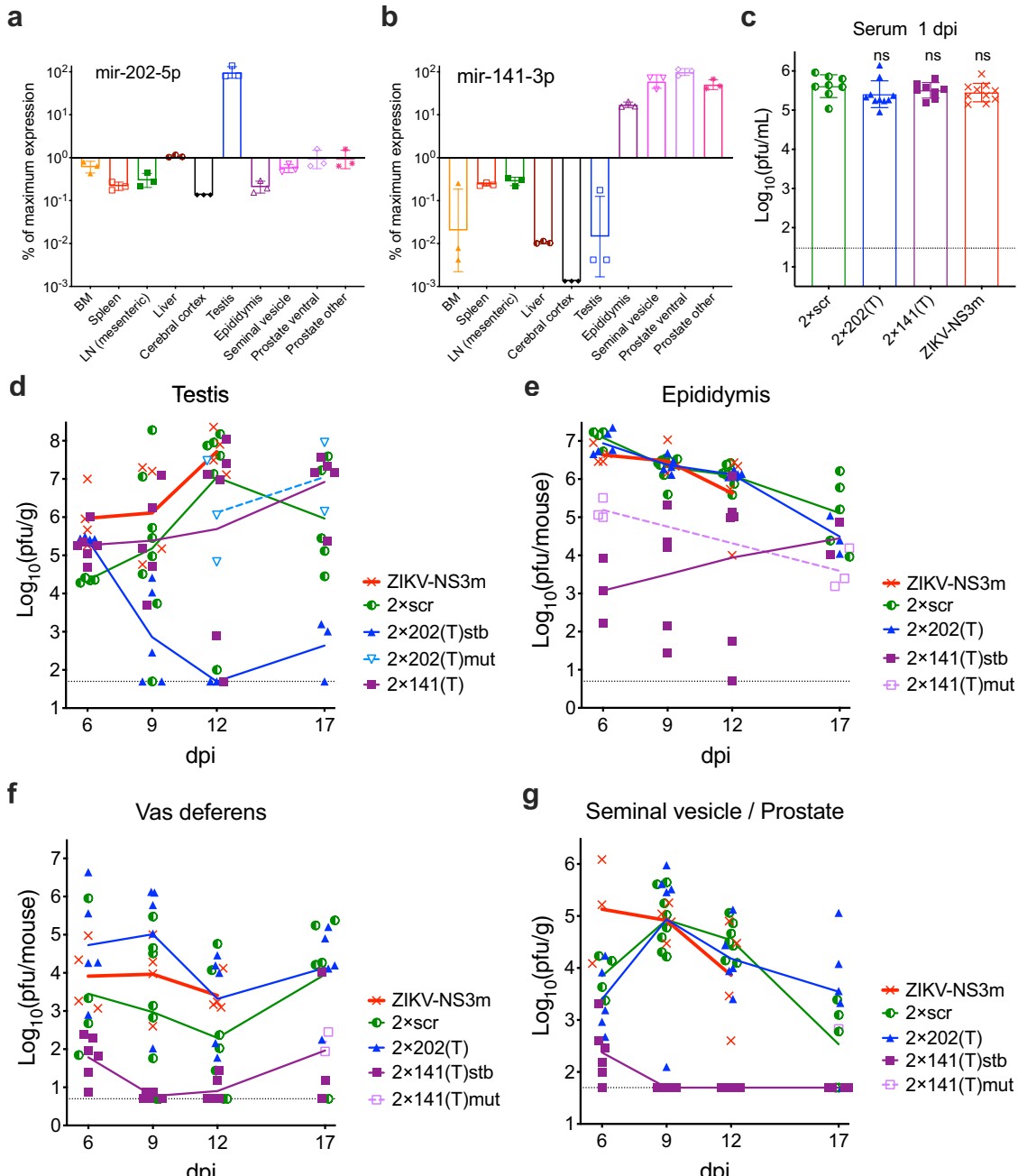

**Fig. 1 Effect of miRNA target insertion on ZIKV replication in the MRS organs of adult AG129 mice.** Relative expression of the mir-202 (**a**) and mir-141 (**b**) in the selected rodent organs based on published miRNA array data[40]. Plotted are the ratios of signal intensity for each miRNA in indicated organs ($n = 3$) normalized to the signal intensity in the testis (**a**) or ventral prostate (**b**) (the organs with a highest expression of corresponding miRNAs). BM bone marrow, LN lymph node. Data are presented as geomegtic mean values ± geomegtic standard deviation. (**c**–**g**) Adult male AG129 mice were infected IP with $10^6$ pfu of ZIKV-NS3m, 2 × scr, 2 × 202(T), or 2 × 141(T) virus. Replication kinetics of indicated viruses in the serum (**c**) and components of the MRS (**d**–**g**). Viruses with stable miRNA-target sequences are indicated as stable (stb). Viruses with deletions or point mutations in the miRNA-target sequences are indicated as mutant (mut). If all virus samples from the organ remained stable, no additional notations were made to the virus name. **c** Data are presented as mean titer ± SD. Differences between the titer of 2 × scr ($n = 8$) and the titer of 2 × 202(T) ($n = 10$), 2 × 141(T) ($n = 8$), and ZIKV-NS3m ($n = 10$) in mouse serum at 1 dpi were compared using one-way ANOVA with multiple comparison adjustment. $p$ values (from left to right): ns, $p = 0.279$; ns, $p = 0.774$; ns, $p = 0.453$. **d**–**g** Data show each replicates (the virus titer in the organ dissected from an individual mouse) and mean virus titers (connecting lines). The limit of detection (dashed lines): 1.5 log$_{10}$(pfu/mL) for serum, 1.7 log$_{10}$(pfu/g) for testis and SV/P, and 0.7 log$_{10}$(pfu/mouse) for epididymis and vas deferens.

compared to 2 × scr. In contrast, accumulation of 2 × 202(T) in the epididymis of mice was comparable to that of 2 × scr, and 2 × 202(T) escape mutants were not detected in this organ. These results support the previous findings suggesting that ZIKV infection of the epididymis can occur via a hematogenous route

independently of the testicular infection[22]. The titer of 2 × 141(T) stb in the epididymis was lowest at 6 dpi, but gradually increased from 9 to 17 dpi. Importantly, escape mutants of the 2 × 141(T) virus [2 × 141(T)mut in Fig. 1e] were detected only in the epididymis, but not in the testis of infected mice. The highest level

of 2 × scr replication in the epididymis occurred ~6 days earlier than in the testis (Fig. 1d, e). Since targeting of ZIKV genome for mir-141 does not block ZIKV replication in the testis (Fig. 1d), this suggests that accumulation of 2 × 141(T)stb in the epididymis between 9 and 17 dpi occurs due to the excurrent transport of the 2 × 141(T) from the testis (Fig. 1e)[22]. Escape mutations were not detected in the 2 × scr virus isolated from the testis and epididymis of any infected mice. Also, instability was not observed in the genome of 2 × 202(T) virus isolated from epididymis—the organ where mir-202 is not expressed (Fig. 1a). Similarly, mir-141 is poorly expressed in the testis (Fig. 1b), and all 2 × 141(T) genomes isolated form this organ remained stable (Fig. 1d). Together, these data demonstrate that escape mutations in the genome of 2 × 202(T) and 2 × 141(T) viruses are selected in a tissue-specific manner, suggesting that this phenomenon can be used to trace the tissue origin of ZIKV genomes transmitted from male to female mouse.

In the vas deferens, ZIKV can either be generated in situ or undergo transit with the flow of infected sperm from the 'upstream' organs (i.e. testis and epididymis, Fig. S3). Between 6 and 17 dpi, replication of only 2 × 141(T)stb in this organ was drastically inhibited. Escape mutants were detected only in mice infected with 2 × 141(T), but not with 2 × 202(T) virus (Fig. 1f), excluding testis as a likely source of the ZIKV in vas deferens. To identify the cell types supporting ZIKV replication in the epididymis and vas deferens, we compared distribution of ZIKV antigens by immunohistochemistry in mice infected with 2 × scr or 2 × 141(T) (Fig. 2). ZIKV-immunoreactivity (ZIKV-IR) was not detected in epithelial cells of vas deferens in any of the mice infected with either virus (Fig. 2e, f; $n = 4$ mice). In contrast, seven out of eight epididymides (87.5%) collected from four mice infected with 2 × scr virus were strongly ZIKV-positive (Fig. 2b, m). This suggests that the high titer of 2 × scr in vas deferens (Fig. 1f) likely corresponds to the intraluminal virus derived from the epididymis. This is in agreement with sequencing results of 2 × 141(T) virus isolated from the different parts of the MRS at 17 dpi (Table S1). For instance, identical escape mutants were detected in the vas deferens and epididymis of two mice (mouse #3 and #4), whilst the virus isolated from the testis of these mice contained intact sequence of miRNA targets. ZIKV-IR was not detected in 87.5% of epididymides (7 out of 8 organs) collected from 4 mice infected with 2 × 141(T) at 9 dpi (Fig. 2c, m). In addition, growth of 2 × 141(T) virus in immortalized mouse distal caput epididymal epithelial cell line DC2 was significantly attenuated compared to 2 × scr virus (Fig. S4), confirming that mir-141 targeting strongly attenuates ZIKV infectivity for epididymal epithelium[22]. Detection of the ZIKV-IR unilaterally in the epididymis of one mouse (Fig. 2m, S5) likely corresponds to the emergence of an escape mutation in the mir-141-targeted virus under miRNA-mediated pressure.

Interestingly, miRNA targets in the 2 × 141(T) virus isolated from the testis, epididymis, and vas deferens of mouse #5 at 17 dpi remained stable (Table S1). This suggests that the ZIKV virus generated in the testis may also reach vas deferens with the efferent flow of the infected sperm. However, the escape mutations in the genome of the 2 × 202(T) virus, which were observed in the testis at 12 and 17 dpi [see 2 × 202(T)mut in Fig. 1d], were not detected in the epididymis and vas deferens of these mice (Fig. 1e, f). This implies that between 6 and 17 dpi trans-epididymal transport of ZIKV from the testis to vas deferens occurs quite inefficiently (as compared to the transport of the virus produced by epididymal epithelial cells), since this transport can be detected only when ZIKV infection of the epididymis is inhibited by mir-141 targeting of the viral genome.

Both 2 × scr and 2 × 202(T) replicated in the SV/P of AG129 mice, reaching peak titer at 9 dpi (Fig. 1g). We observed that

2 × scr virus primarily targets epithelial cells of the seminal vesicle (Fig. 2h, m); however, infection of the epithelial cells of the prostate occurred less frequently (Fig. 2k, m). ZIKV genome targeting for mir-141 effectively blocks infection of epithelial cells (Fig. 2i, l, m) and restricts virus replication in both organs (Fig. 1g). Escape mutations were detected in the SV/P at 17 dpi only in one out of five mice (see mouse #3) infected with 2 × 141(T) virus. The same escape mutants were also detected in the epididymis and vas deferens, but not in the testis of the mouse #3 (Table S1). This suggests that these escape mutants originated in the epididymis and were transported to the SV/P via vas deferens. Since the titer of the escape mutant in the epididymis of mouse #3 was considerably higher than that in the SV/P (mass of epididymis is ~0.1 g), a reverse spread of the mutant virus from the SV/P to the epididymis against the excurrent flow of sperm seems unlikely. This indicates that ZIKV escape mutant detected in the SV/P might be either of a local origin, or it could have been generated in the epididymis.

Together, these results indicate that with the exception of the testis, the epithelial cells expressing mir-141 are a common cell type capable of supporting productive ZIKV replication in the major MRS tissue components. Importantly, ZIKV infection of these cells can be restricted by mir-141 targeting of viral genome.

**ST of miRNA targeted ZIKV from male to female mice between 7 and 17 dpi.** Next, we evaluated the effect of miRNA targeting of ZIKV genome on the efficiency of viral transmission from male to female AG129 mice. We focused on the events which occur between 7 and 17 days post male infection. This interval coincides with peak of ZIKV shedding into the semen of immunodeficient mice [9–11 dpi[19]] and humans [7–11 days post onset of ZIKV symptoms[49]]. Male mice were infected IP with $10^6$ pfu of 2 × scr, 2 × 202(T), or 2 × 141(T) virus (Fig. 3a). At 7 dpi, each infected male was placed into a separate cage with a non-infected female. Mice were allowed to mate for 10 days, during which time females were periodically bled to perform virus isolation. Male mice were sacrificed at 17 dpi and tissues were collected. A female mouse was considered infected if (i) infectious ZIKV was detected in the serum (or in the brain if death occurred) by titration in Vero cells; and/or (ii) ZIKV-specific neutralizing antibodies (NA) were detected in the serum at 28 days post mating using 50% plaque reduction neutralization assay ($PRNT_{50}$).

Compared to 2 × scr, targeting of ZIKV genome for testis-specific mir-202 had no effect on the efficiency of ZIKV ST (Fig. 3b, $p > 0.05$; Fishers' exact test). In contrast, the transmission rate of the 2 × 141(T) virus was significantly reduced compared to 2 × scr (Fig. 3b, $p < 0.001$; Fishers' exact test). The efficiency of ZIKV male-to-female ST may depend on: (i) a dose of the virus delivered into female reproductive system (FRS) with the semen; (ii) virulence of native or mutated virus for the FRS; and (iii) susceptibility of the FRS to infection. Therefore, to determine which of these factors could explain the observed reduction in the efficiency of ST of the mir-141-targeted virus, we performed a separate experiment using a direct intravaginal inoculation of the female AG129 mice with the $10^5$ pfu of the 2 × scr, 2 × 202(T), or 2 × 141(T) virus. The same criteria of virus infectivity to female mouse (see above) were applied to determine whether a female mouse became infected. We observed that intravaginal infectivity of all three viruses was similar (90%) (Fig. 3c), suggesting that neither virulence of the virus nor susceptibility of the FRS to infection were the determining factors influencing efficiency of ZIKV ST. These results also suggest that low efficiency of ST of 2 × 141(T) virus is likely due to insufficient shedding of the virus into the semen.

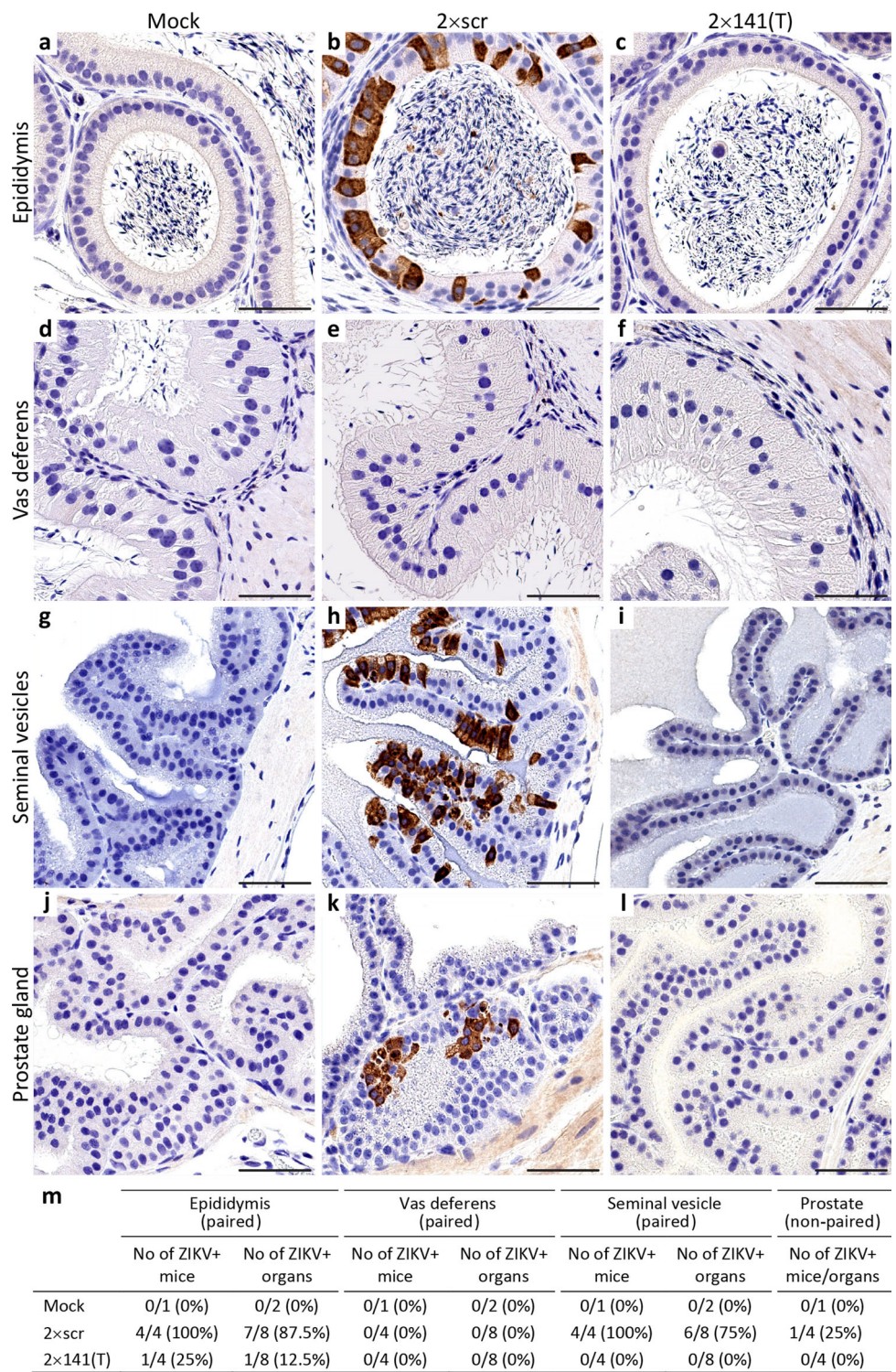

**Fig. 2 Genome targeting for mir-141 restricts ZIKV ability to infect epithelial cells of epididymis, seminal vesicles, and prostate.** Adult AG129 mice were mock-inoculated (**a**, **d**, **g**, **j**; $n = 1$) or infected IP with $10^6$ pfu of $2 \times$ scr (**b**, **e**, **h**, **k**; $n = 4$) or $2 \times 141$(T) (**c**, **f**, **i**, **l**; $n = 4$) virus. Representative images of immunoreactivity (IR) for ZIKV antigen in the epididymis (**a–c**), vas deferens (**d–f**), seminal vesicles (**g–i**), and prostate (**j–l**) at 9 dpi (experiment was performed once). Scale bars: 50 µm. **m** Summary of ZIKV-IR detection in the MRS of mice infected with $2 \times$ scr or $2 \times 141$(T) viruses as compared to mock. A paired mouse organs (epididymis, vas deferens, and seminal vesicles) were considered ZIKV positive (ZIKV+ mouse), if ZIKV+ cells were detected either unilaterally or bilaterally.

To establish the organ of origin of sexually transmitted virions within the MRS, we isolated viruses from the female serum (Fig. 3a) and compared viral sequences with those isolated from the MRS of the males involved in the mating. Scr sequences of $2 \times$ scr remained stable in all samples (female serum and MRS organs) isolated from mice. All $2 \times 202$(T) viruses isolated from the serum of sexually infected females ($n = 9$) also contained intact miRNA targets (Fig. 3d). In contrast, only four male mating partners for those females ($n = 9$) contained stable $2 \times 202$(T) viruses in their testes at 17 dpi, and mean viral load in these testes

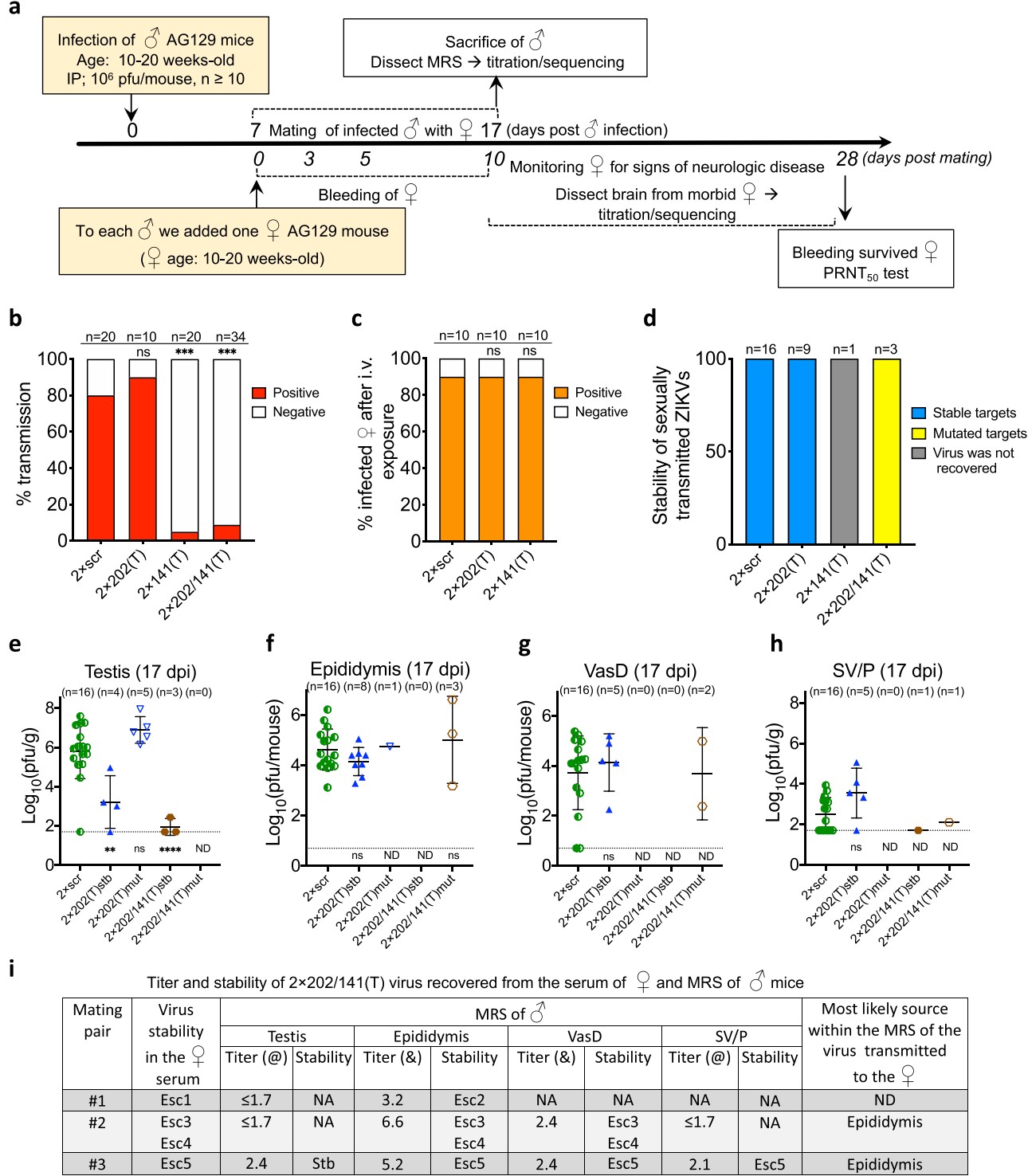

**i** Titer and stability of 2×202/141(T) virus recovered from the serum of ♀ and MRS of ♂ mice

| Mating pair | Virus stability in the ♀ serum | MRS of ♂ | | | | | | | | Most likely source within the MRS of the virus transmitted to the ♀ |
|---|---|---|---|---|---|---|---|---|---|---|
| | | Testis | | Epididymis | | VasD | | SV/P | | |
| | | Titer (@) | Stability | Titer (&) | Stability | Titer (&) | Stability | Titer (@) | Stability | |
| #1 | Esc1 | ≤1.7 | NA | 3.2 | Esc2 | NA | NA | NA | NA | ND |
| #2 | Esc3 Esc4 | ≤1.7 | NA | 6.6 | Esc3 Esc4 | 2.4 | Esc3 Esc4 | ≤1.7 | NA | Epididymis |
| #3 | Esc5 | 2.4 | Stb | 5.2 | Esc5 | 2.4 | Esc5 | 2.1 | Esc5 | Epididymis |

was significantly lower than that of 2 × scr virus (see 2 × 202(T) stb in Fig. 3e, p < 0.001, one-way ANOVA). In the testes obtained from the remaining five mice, the 2 × 202(T) virus acquired escape mutations (see 2 × 202(T)mut in Fig. 3e), which were associated with ~5000-fold increase in the virus titer in the testis as compared to the titer of 2 × 202(T)stb. However, these escape mutants were not detected in the female serum (Fig. 3d), which suggests that testicular infection of ZIKV does not contribute to the pool of the viruses transmitted from male to female mice.

At 17 dpi, the titers of 2 × 202(T)stb and 2 × scr viruses were very similar in the epididymis, vas deferens, and SV/P (Fig. 3f–h),

suggesting that all these organs can be a source of 2 × 202(T)stb virus that was detected in the female serum (Fig. 3d). In contrast, the 2 × 141(T) virus was attenuated in these components of MRS (Fig. 1d–g). Only a single female (5%; n = 20) developed a detectable anti-ZIKV NA titer after mating with male infected with 2 × 141(T) virus (Fig. 3b, d). However, infectious 2 × 141(T) virus was not detected in the serum nor the brain of this animal, precluding subsequent analysis of the origin of the transmitted 2 × 141(T) virus in the MRS.

We also analyzed ST efficiency of the 2 × 202/141(T) virus (Fig. S2)[22], which contains two target sequences for both mir-202 and

**Fig. 3 ST of miRNA-targeted ZIKVs from male to female mice. a** Design of the ZIKV ST experiment. **b** ST rate (%) of the 2 × scr and miRNA-targeted ZIKVs. **c** Infection rate (%) of indicated viruses in AG129 female mice following intravaginal inoculation with the dose of $10^5$ pfu of each virus. Difference in the ST rates (**b**) and in the intravaginal infection rates (**c**) between 2 × scr and miRNA-targeted viruses was compared using two-tailed Fisher's exact test followed by Bonferroni correction for multiple comparisons (ns, $p > 0.999$). **d** Stability of miRNA targets or scr sequences in ZIKV genomes recovered from the serum of female mice following male-to-female sexual virus transmission. **e–h** Mean titer ± standard deviation (SD) at 17 dpi and miRNA target stability of ZIKVs recovered from the testis (**e**), epididymis (**f**), vas deferens (VasD) (**g**), and SV/P (**h**) of male mice participating in the mating experiment. Reproductive organs (number is shown in each panel) were analyzed only for those males whose female mating partners developed detectable ZIKV viremia. (Note: vas deferens and SV/P were not collected from 4 out of 9 male mice infected with 2 × 202(T) and from 1 out of 3 mice infected with 2 × 202/141(T) viruses). The dashed lines indicate the limit of virus detection: 1.7 $\log_{10}$(pfu/g) for testis and SV/P (**e, h**) and 0.7 $\log_{10}$(pfu/mouse) for epididymis and vas deferens (**f, g**). **i** Sequence comparison of the 2 × 202/141(T) virus recovered from female serum and from different organs within the MRS of the corresponding mating male partner. Esc escape mutant, @ virus titer in the testis and SV/P is expressed as $\log_{10}$(pfu/g), & virus titer in the epididymis and vas deferens is expressed as $\log_{10}$(pfu/mouse). NA—virus titer or target stability was not assessed. Differences between the titers of 2 × scr and each of the miRNA-targeted viruses in the mouse organs were compared using one-way ANOVA with multiple comparison adjustment (Dunnett's test). **$p < 0.01$; ***$p < 0.001$; ****$p < 0.0001$. $p$ values (from left to right): **e** ns, $p = 0.3338$. **f** ns, $p = 0.5073$; ns, $p = 0.8552$. **g** ns, $p = 0.8181$, **h** ns, $p = 0.1116$. ND—$p$-value is not determined.

mir-141. In contrast to the 2 × 202(T) or 2 × 141(T) constructs, presence of targets for both miRNAs in the 2 × 202/141(T) virus genome allows for the independent accumulation of escape mutants in the testis and in the mir-141-expressing epithelial cells within the MRS. Similar to the 2 × 141(T), transmission rate of the 2 × 202/141(T) virus was significantly attenuated (8.8%; $n = 34$) as compared to the 2 × scr (80%; $n = 20$; Fig. 3b, $p < 0.001$; Fishers' exact test). Interestingly, 2 × 202/141(T) viruses isolated from the serum of all mating females contained deletions in the miRNA-targeting region (Fig. 3d, i). Analysis of viruses isolated from the MRS of the male mating partners indicated that it is unlikely that transmitted viruses were generated within the testis. The 2 × 202/141(T) virus isolated from the testis was either stable (see mating pair #3 in Fig. 3i) or the testicular virus titer was below the limit of virus detection (see mating pair #1 and #2 in Fig. 3i), and, therefore, their sequences were not assessed. In addition, an escape mutation in the 2 × 202/141(T) virus, which was isolated from serum of a female from mating pair #3 eliminated both targets for mir-141, but preserved one target for mir-202 (see Esc5 in Figs. 3i, S6). This suggests that replication of the ZIKV with such deletion in the 3'NCR likely remained restricted only in mouse testis, but not in the MRS cells expressing mir-141.

Escape mutants detected in the 2 × 202/141(T) virus isolated from females from mating pairs #2 and #3 matched exactly those found in the epididymis and vas deferens of their mating partners (Fig. 3i). Also, for the male from mating pair #3 the matching deletion was also found in the SV/P sample (see Esc5 in Figs. 3i, S6). Considering that (i) ZIKV infection of the cells of vas deferens occurs very inefficiently (Figs. 1 and 2) and that (ii) ZIKV generated in the epididymis can be transported to the SV/P with the flow of infected sperm through vas deferens (see mouse #3 in Table S1), the epididymis is the most likely source of the viruses transmitted from male to female mice (Fig. 3i).

The deletion in the 2 × 202/141(T) virus genome isolated from the female in mating pair #1 does not match the deletion found in the epididymis of their mating partner (Fig. 3i). It is possible that Esc1 could have been generated in the accessory glands or ducts of the MRS. Regretfully, these parts of the MRS of this animal were not preserved, precluding a more definitive analysis of the origin of the Esc1 mutant. It is also possible that Esc1 could have been generated in the epididymis but was subsequently washed out/replaced by another dominant escape mutant (Esc2) by 17 dpi.

**Analysis of ZIKV shedding into the semen between 7 and 11 dpi.** The low ST rate of viruses containing targets for mir-141 (Fig. 3b) may be attributed to an insufficient shedding of these

viruses into the semen. Alternatively, inefficient ST of the 2 × 141 (T) virus might be attributed to its attenuated vaginal infection and replication in the FRS (Fig. S7). To directly test the effect of mir-141 targeting on ZIKV shedding into the semen, we infected AG129 male mice with 2 × scr or 2 × 141(T) virus, and at 7 dpi each male was placed with two uninfected female CD-1 mice (to increase the probability of mating). Mice were allowed to mate for 4 days (Fig. 4a) instead of 10 days as was done in the study depicted in Fig. 3a (we noticed that most male mice used in that experiment copulated with females no later than 4–5 days of cohabitation). Female CD-1 mice that developed vaginal plugs were sacrificed to determine viral load in the FRS by titration in Vero cells. A mean ZIKV titer was reported if both CD-1 females from the same cage developed vaginal plugs. None of 2 × 141(T)-infected AG129 males ($n = 15$) deposited ZIKV+ semen into FRS of CD-1 females (Fig. 4b). In comparison, 70% of mating males ($n = 10$) infected with 2 × scr virus were able to transmit variable viral loads ($p > 0.001$, 2-tailed Fisher's exact). This experiment reinforces the notion that the cells of epithelial origin in the MRS play a determining role in the sexual transmissibility of the ZIKV, since the blocking of ZIKV replication in such cells by targeting the viral genome for mir-141-mediated degradation efficiently inhibits viral shedding into semen and prevents the ST.

**Tracing individual ZIKV genomes during ST of the barcoded ZIKVs from male to female mice.** Mir-141 miRNA is not confined to the epithelial cells of the MRS, but it is also expressed in a variety of other tissues[40–42]. This suggests that infectivity and titers of the mir-141-targeted ZIKV clones may become reduced before the seeding of the MRS. In turn, this may translate into a diminished shedding of the mir-141-targeted viruses into the ejaculate and result in inefficient ST. Therefore, we sought to apply a different methodology to verify our finding that the ZIKV shedding into the ejaculate and subsequent ST are defined by ZIKV replication in the cells of the epithelial origin in the MRS. One way of doing this is to analyze the genetic bottlenecks imposed onto ZIKV populations passing through the anatomical barriers and/or specific cell types of the MRS[50].

For that, we modified 2 × scr virus to generate a miRNA-target-free library of ZIKV genomes containing a stretch of 11 random nucleotides (designated n11). The n11 sequences partially substitute one of the two scr sequences in the 2 × scr genome (Fig. 5a), which can serve as 'barcode' tags for differentiating viral genomes. A plasmid library consisting of ~9.3 × $10^4$ clones was transfected into Vero cells, generating ZIKV-lib/n11 virus library. Male AG129 mice were infected with ZIKV-lib/n11 viruses followed by a semen shedding experiment as depicted in Fig. 5b. Unlike the study depicted in Fig. 4a, in this experiment male mice

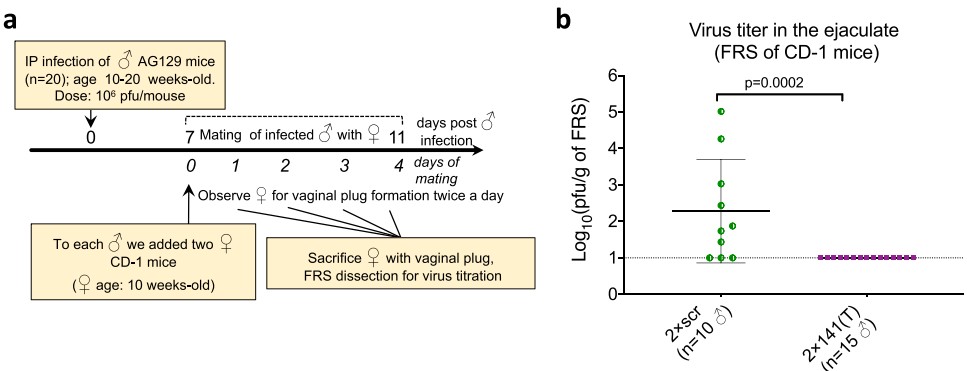

**Fig. 4 Genome targeting for mir-141 restricts ZIKV shedding into the semen between 7 and 11 dpi. a** Experimental design. Male AG129 mice were infected IP with $10^6$ pfu of 2 × scr or 2 × 141(T) virus, and between 7 and 11 dpi mice were allowed to mate with two CD-1 females in the individual cages. **b** Mean virus titer ± SD in the ejaculate of AG129 mice that was deposited into FRS of CD-1 mice. Each dot represents an average ZIKV titer in the FRS of CD-1 females that developed vaginal plug (one or two females per cage) after mating with ZIKV-infected male partner. The dashed line indicates the limit of virus detection: 1.0 $\log_{10}$(pfu/g of FRS). Differences between viral titers in the FRS were compared using two-tailed Mann–Whitney test.

were sacrificed immediately after the vaginal plug was detected in one of the two mating partners. This modification allows to minimize the time interval between semen deposition into FRS and the MRS collection. The MRS was also collected from several males who did not deposit semen to female partners by 11 dpi.

We reasoned that similar to a simian immunodeficiency virus (SIV) infection of MRS in cynomolgus macaques[51], the hematogenous dissemination of ZIKV from the infectious site should be followed by virus seeding and replication in the organs of the MRS[22,24]. This process is associated with genetic bottlenecks imposed by anatomical barriers and/or changes in the target cell types. This should translate into a reduction of a diverse ZIKV-lib/n11 population and production of a tissue-dependent barcode composition of virus progenies (Fig. 5c). Mouse organs that are not connected by known anatomical routes of virus migration (i.e. brain and MRS) should not have overlapping sets of barcodes. In contrast, since ZIKV can be transported within the MRS by the flow of sperm[22] containing virus and/or with secretions of accessory glands, a considerable similarity in the barcode signatures is expected in the ZIKV isolated from different parts of the MRS. The barcode composition of the virus that is sexually transmitted from infected MRS with the ejaculate should be determined by the compositions of the virus outputs from the anatomical components participating in the ejaculation (Fig. 5c).

We deep-sequenced ZIKV-lib/n11 viruses isolated from serum, brain, and organs of MRS of seven male mice participating in the mating experiment. Four of these mice engaged in coitus (which coincidentally occurred at 10 dpi) and all of them deposited ZIKV + semen into FRS. The other three male mice were randomly selected from mice sacrificed at 11 dpi, which did not engage in coitus. In our analysis, we focused only on the most represented barcodes, which together constitute 90% of total barcode reads (reads$_{90}$) and ignored the remaining underrepresented barcodes. This minimizes the effect of mutations occurring during sample preparation on barcode heterogeneity. It also eliminates non-replicating genomes (background), which could have been seeded in the organ during initial ZIKV-lib/n11 dissemination from the site of infection. Compared to the inoculum, the heterogeneity of barcoded viruses in the mouse serum at 1 dpi was reduced by ~7-fold (Fig. 5d). In turn, there was further 100–1000-fold reduction in a number of barcodes in reads$_{90}$ between the serum and different mouse organs, suggesting that ZIKV encounters strong genetic bottlenecks during dissemination from the site of infection into mouse organs.

Next, we tested whether ZIKV-lib/n11 viruses can be used to identify previously established ZIKV migration routes from the testis to epididymis[22]. For that, we identified all barcodes in reads$_{90}$ that are common between viruses in the testis and the organ in question, and calculated combined frequencies of common barcodes (CFCB) among all barcoded viruses which were found in these two particular mouse organs. There was almost no similarity among barcodes in reads$_{90}$ detected in the brain and testis (Fig. 5e), reflecting independent ZIKV infection of these organs from a pool of viruses with a high heterogeneity of n11 region. In contrast, mean CFCB value for the testis–epididymis pair was 58.2%, which was significantly higher than that of the brain–testis pair (Fig. 5e, $p < 0.0001$). This strongly suggests ZIKV migration between these two organs. Importantly, there were no significant differences between the CFCB values for viruses detected in the testis–vas deferens or testis–SV/P pairs, as compared to the brain–testis pair (Fig. 5e; $p > 0.999$ and $p = 0.622$, respectively). The absence of a high similarity in barcodes between the testis and vas deferens at 11 dpi suggests that ZIKV generated in the testis has not yet completed a full transition through the epididymis to be expelled into the vas deferens. These findings are in agreement with the kinetics of 2 × 141(T)stb accumulation in the MRS (Fig. 1d–f) showing that between 6 and 12 dpi, testis-derived 2 × 141(T)stb is present in the epididymis but not in the vas deferens.

Next, we analyzed whether ZIKV generated in the epididymis can be detected in the downstream MRS organs. Almost no similarity was detected among barcodes in reads$_{90}$ isolated from the epididymis and brain (Fig. 5f). In contrast, mean CFCB value for the epididymis–vas deferens pair was 51.1% (Fig. 5f), which was significantly higher compared to the CFCB values for the epididymis–brain or epididymis–SV/P pair (Fig. 5f). Perhaps not surprisingly, these results are in line with the MRS physiology and our earlier detection of the epididymis-generated miRNA-targeted ZIKV in the vas deferens (Fig. 3i, Table S1).

Next, we asked whether the barcode compositions of the viruses detected in the major anatomical component of the MRS would allow identification of the tissue/cellular sources of sexually transmitted ZIKV. The n11 region of ZIKV-lib/n11 viruses in the ejaculate that was recovered from the FRS of CD-1 mice ($n = 4$) after mating (Fig. 5b) was sequenced and compared to the n11 region of ZIKV-lib/n11 viruses isolated from the MRS organs. None of the seven dominant barcodes present in the ejaculate could be traced back to the testis (not detected in the reads$_{90}$ and frequency < 0.1%) (Fig. 6). In contrast, all seven dominant

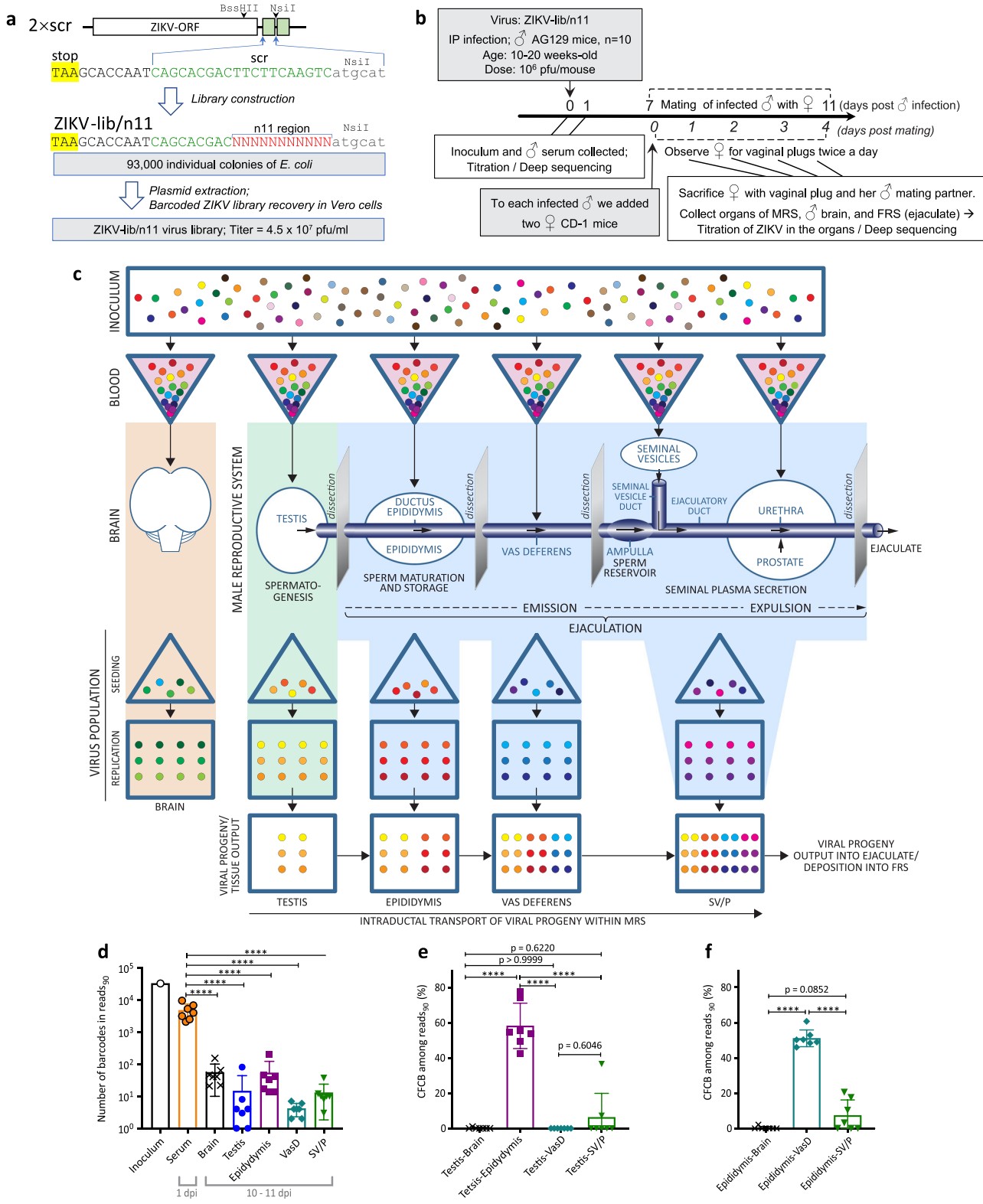

barcodes present in the ejaculate were detected in the epididymis (detected in the reads$_{90}$ at a frequency of 0.7–6.8%). Some of these barcodes were also detected at high frequencies in the vas deferens and SV/P, most likely reflecting the excurrent transit and potential seeding of these downstream components of the MRS (see Fig. 5c for the anatomical/physiological schematic) with the sperm and/or epididymal secretions. Together, these findings identify the epididymis (specifically, the epididymal epithelial cells) as the main contributor of ZIKV output to the ejaculate and subsequent ST of the virus during the early course of infection (10 dpi).

**Restrained male-to-female ST of live-attenuated ZIKV vaccine candidate**. We previously reported the construction of a live-attenuated ZIKV vaccine candidate virus C/3′NCR-mir(T) that

**Fig. 5 Construction and application of a library of ZIKV genomes with unique molecular identifiers (barcodes). a** Construction of a barcoded library of ZIKV genomes using genetic background of 2 × scr virus. Green boxes and letters indicate scr sequences. Red letters highlight a stretch of 11 random nucleotides. **b** Experimental design of a semen shedding study using AG129 mice infected with a barcoded ZIKV library. **c** Hypothetical reduction of barcoded virus diversity during hematogenous dissemination of the ZIKV-lib/n11 and shedding of the viruses into FRS with ejaculate. **d** Mean number of barcodes in the reads90 ± SD of ZIKV-lib/n11 virus population isolated from: mouse serum at 1 dpi ($n = 7$); mouse organs at 10–11 dpi ($n = 7$); or from the inoculum that was used for mouse infection ($n = 1$). Difference between the number of barcodes in reads90 in the serum and mouse organs was compared using one-way ANOVA with multiple comparison adjustment (Dunnett's test). **e, f** Mean combined frequencies of common barcodes (CFCB) among reads90 (%) ± SD for ZIKV-lib/n11 viruses recovered from the testis (**e**) or epididymis (**f**) and various organs of AG129 male mice ($n = 7$). Differences between common barcode frequencies among organ pairs were compared using one-way ANOVA with multiple comparison adjustment (Tuke's test). ****$p < 0.0001$.

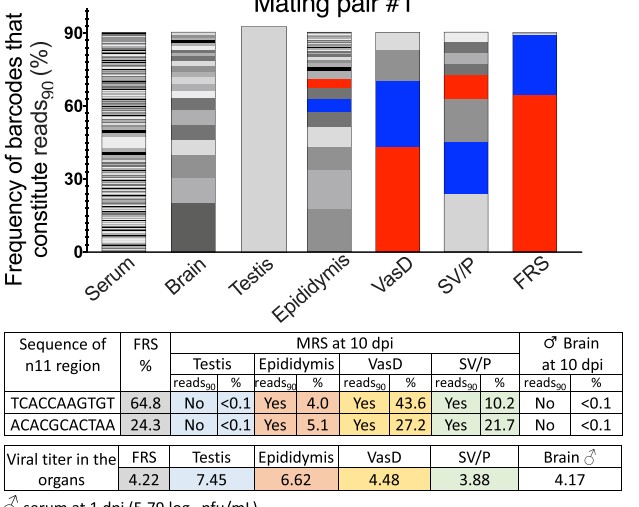

## Mating pair #1

| Sequence of n11 region | FRS % | MRS at 10 dpi | | | | | | | | ♂ Brain at 10 dpi | |
|---|---|---|---|---|---|---|---|---|---|---|---|
| | | Testis | | Epididymis | | VasD | | SV/P | | | |
| | | reads90 | % | reads90 | % | reads90 | % | reads90 | % | reads90 | % |
| TCACCAAGTGT | 64.8 | No | <0.1 | Yes | 4.0 | Yes | 43.6 | Yes | 10.2 | No | <0.1 |
| ACACGCACTAA | 24.3 | No | <0.1 | Yes | 5.1 | Yes | 27.2 | Yes | 21.7 | No | <0.1 |

| Viral titer in the organs | FRS | Testis | Epididymis | VasD | SV/P | Brain ♂ |
|---|---|---|---|---|---|---|
| | 4.22 | 7.45 | 6.62 | 4.48 | 3.88 | 4.17 |

♂ serum at 1 dpi (5.79 log10pfu/mL)

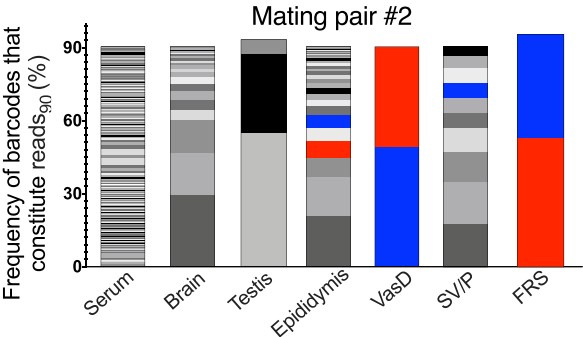

## Mating pair #2

| Sequence of n11 region | FRS % | MRS at 10 dpi | | | | | | | | ♂ Brain at 10 dpi | |
|---|---|---|---|---|---|---|---|---|---|---|---|
| | | Testis | | Epididymis | | VasD | | SV/P | | | |
| | | reads90 | % | reads90 | % | reads90 | % | reads90 | % | reads90 | % |
| TACCCCAGCAC | 53.1 | No | <0.1 | Yes | 6.8 | Yes | 42.6 | No | <0.1 | No | <0.1 |
| GTATCCACACG | 42.3 | No | <0.1 | Yes | 5.0 | Yes | 51.4 | Yes | 6.2 | No | <0.1 |

| Viral titer in the organs | FRS | Testis | Epididymis | VasD | SV/P | Brain ♂ |
|---|---|---|---|---|---|---|
| | 5.44 | 8.13 | 6.50 | 4.12 | 3.54 | 5.21 |

♂ serum at 1 dpi (5.68 log10pfu/mL)

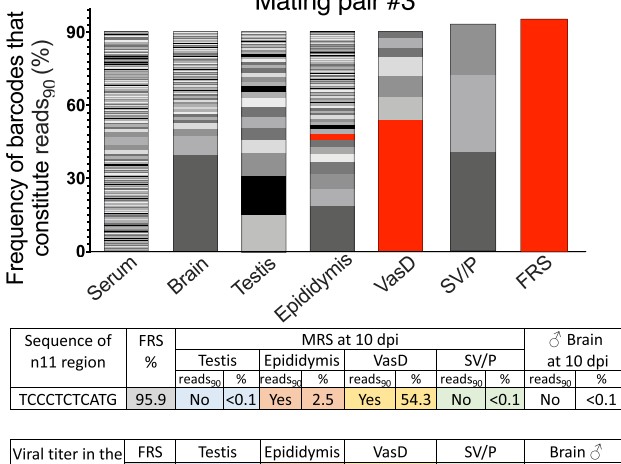

## Mating pair #3

| Sequence of n11 region | FRS % | MRS at 10 dpi | | | | | | | | ♂ Brain at 10 dpi | |
|---|---|---|---|---|---|---|---|---|---|---|---|
| | | Testis | | Epididymis | | VasD | | SV/P | | | |
| | | reads90 | % | reads90 | % | reads90 | % | reads90 | % | reads90 | % |
| TCCCTCTCATG | 95.9 | No | <0.1 | Yes | 2.5 | Yes | 54.3 | No | <0.1 | No | <0.1 |

| Viral titer in the organs | FRS | Testis | Epididymis | VasD | SV/P | Brain ♂ |
|---|---|---|---|---|---|---|
| | 4.10 | 7.93 | 6.10 | 3.48 | 3.21 | 5.98 |

♂ serum at 1 dpi (5.85 log10pfu/mL)

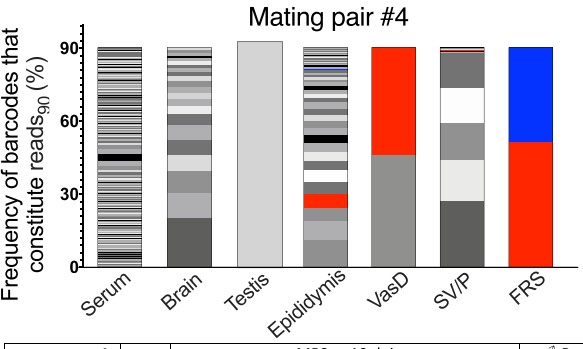

## Mating pair #4

| Sequence of n11 region | FRS % | MRS at 10 dpi | | | | | | | | ♂ Brain at 10 dpi | |
|---|---|---|---|---|---|---|---|---|---|---|---|
| | | Testis | | Epididymis | | VasD | | SV/P | | | |
| | | reads90 | % | reads90 | % | reads90 | % | reads90 | % | reads90 | % |
| GGCCAGTAGCC | 51.8 | No | <0.1 | Yes | 5.4 | Yes | 45.0 | Yes | <0.1 | No | <0.1 |
| CTGTCAGACAA | 38.8 | No | <0.1 | Yes | 0.7 | No | <0.1 | Yes | <0.1 | No | <0.1 |

| Viral titer in the organs | FRS | Testis | Epididymis | VasD | SV/P | Brain ♂ |
|---|---|---|---|---|---|---|
| | 1.30 | 7. 60 | 5.88 | 1.18 | 4.09 | 4.26 |

♂ serum at 1 dpi (5.73 log10pfu/mL)

**Fig. 6 Tracing individual ZIKV genomes during male to female ST of ZIKV-lib/n11 viruses.** Four male AG129 mice infected with ZIKV-lib/n11 deposited ZIKV+ ejaculate into FRS of CD-1 mice at 10 dpi (see Fig. 5b). For each mating pair, a top panel shows the frequency of barcodes that together constitute reads90 for the individual male or female organ. Each color represents a unique barcode. Only barcodes with frequencies of >1% in the FRS are shown as blue or red colors. Sequences and frequencies of these barcodes are presented in the middle panels. "Yes" or "No" denote whether the barcodes were among the reads90 for the given organ. Bottom panels for each mating pair show the titers of ZIKV-lib/n11 viruses in mouse organs collected at 10 dpi. Virus titers in the FRS (ejaculate), testis, SV/P, and brain of males are expressed as log10(pfu/g). Virus titers in the epididymis and vas deferens are expressed as log10(pfu/mouse).

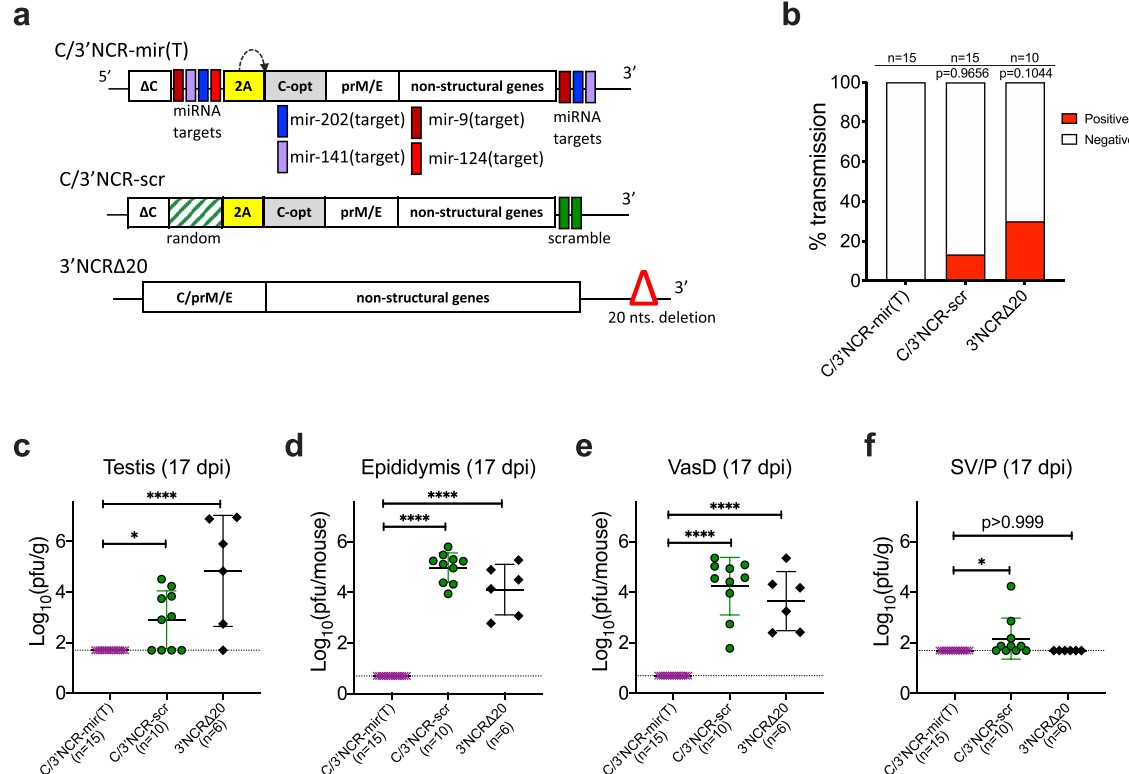

**Fig. 7 Restriction of male-to-female transmissibility of the miRNA-targeted live-attenuated ZIKV vaccine candidate. a** Schematic representation of viral genomes used in the study. **b** Rate of ST (%) of indicated viruses (determined as in Fig. 3a). Differences in the rates of ST between C/3′NCR-mir(T) and control viruses were compared by two-tailed Fisher's exact test with Bonferroni correction. **c–f** At 17 dpi, organs of MRS were dissected from mice participating in the ST experiment. Graphs show mean virus titer ± SD in the organs of AG129 mice. The dashed lines indicate the limit of virus detection: 1.7 $\log_{10}$ pfu/g for testis and SV/P and 0.7 $\log_{10}$ pfu/mouse for epididymis and vas deferens. Differences between viral titers in the organs of MRS were compared using one-way ANOVA with multiple comparison adjustment (Dunnett's test). *$p < 0.05$; ****$p < 0.0001$.

contains multiple miRNA targets (Fig. 7a), including two targets for the testicular mir-202 miRNA, two targets for the epithelial mir-141miRNA, and targets for mir-9 and mir-124 miRNAs that are highly expressed in the central nervous system (CNS). These targets are inserted into a duplicated C-gene region (dCGR) and 3′NCR of the parental ZIKV-NS3m virus genome. To assure that incorporation of these miRNA targets eliminates any potential of the vaccine candidate to be sexually transmitted, we compared the male-to-female transmission rate of the C/3′NCR-mir(T) with that of C/3′NCR-scr and 3′NCRΔ20 viruses (Fig. 7a) using the experimental design depicted in Fig. 3a. Both C/3′NCR-scr and 3′NCRΔ20 viruses (Fig. 7a) were constructed using the genetic background of ZIKV-NS3m[22] and were used here as controls; C/3′NCR-scr contains scr and other random sequences inserted into the dCGR and 3′NCR, while 3′NCRΔ20 is attenuated by a 20 nt deletion in the 3′NCR[22,52,53], a mechanism of attenuation that is different from the miRNA targeting.

We report a zero ST rate for the vaccine candidate C/3′NCR-mir(T) ($n = 15$ mating pairs), while the control viruses were sexually transmissible, although at the rate which was lower than that of 2 × scr virus (Fig. 7b). Furthermore, the C/3′NCR-mir(T) virus was undetectable in the MRS tissues at 17 dpi (Fig. 7c–e). This is in contrast to the control viruses, both of which produced relatively high virus loads in the testis, epididymis, vas deferens, and less so in the accessory glands. These results support the strategy for using miRNA targeting to inhibit ZIKV tropism for the MRS, thereby eliminating any potential of the virus to be sexually transmitted. This reinforces the safety of the C/3′NCR-mir(T) vaccine candidate virus.

## Discussion

To delineate the time frame when male mice would have the highest potential for ST of genetically modified ZIKV clones, we analyzed the infection events/kinetics occurring in the MRS of adult AG129 mice from 6 to 17 dpi. The control 2 × scr virus reaches the peak of its replication in the testis at 12 dpi, in the epididymis at 6 dpi, and in the SV/P at 9 dpi, followed by a decline of viral titers in all these MRS components by 17 dpi (Fig. 1). This suggests that ZIKV virions with highest potential for ST are likely produced in the MRS prior to 17 dpi, which justifies our decision to terminate ST studies at 17 dpi (Fig. 3a). Subsequently, we observed that 2 × scr virus transmission to female AG129 mice occurs mostly before 12 days post male mouse infection. This allowed for a considerable reduction of the mating time for semen shedding studies [7–11 days post male mouse infection (Figs. 4a, 5b)]. This particular interval (7–11 dpi) coincides with the peak of virus shedding into the semen of immunodeficient mice [7–12 dpi[19]] and human semen [7–11 days post the onset of ZIKV symptoms observed for natural isolates of ZIKV[49]].

It is important to note that shedding of infectious ZIKV virions into mouse semen does not stop abruptly at 11 dpi, but continues up to 22 dpi, although at a much lower rate[19]. This suggests that additional studies may be needed to elucidate the source of infectious ZIKV that is sexually transmitted during the late phase of viral infection (12–21 dpi), as well as the source of non-infectious ZIKV RNA, which can persist in mouse and human semen for months after the onset of ZIKV symptoms[13,19,49]. There are, however, several limitations of our experimental model

**Role of the epididymal epithelium as a source of the ZIKV virions/genomes secreted into the ejaculate**. The results of two independent yet complementary approaches used in this study identified the epididymis as a leading source of the sexually transmitted ZIKV genomes during the early phase of infection (7–11 dpi). This conclusion is based on the several lines of evidence. First, the epithelial cells of vas deferens were refractory to ZIKV infection (Fig. 2e, m), making the vas deferens an unlikely source of the de-novo produced ZIKV virions shedded into the ejaculate. Second, at the time of virus shedding into the ejaculate and sexual transmission (9–11 dpi), most of ZIKV virions were likely generated by the epididymis, but not testis or SV/P. This is consistent with our observations that the peak titer of 2 × scr virus in the epididymis was ~300 fold higher than that in the SV/P, when normalized by mass (Fig. 1e, g). Furthermore, the peak of 2 × scr replication in the epididymis was reached at least 6 days earlier than that in the testis and 3 days earlier than that in the SV/P (Fig. 1). Third, the restriction of ZIKV replication in the epididymal epithelium by mir-141 targeting of viral genome (ref. [22] and Figs. 1e, 2c, 2m in this study) correlates with an attenuated transmission of 2 × 141(T) and 2 × 202/141(T) viruses from male to female AG129 mice (Fig. 3b), which is also in agreement with a reduced shedding of the 2 × 141(T) virus into the ejaculate (Fig. 4b). Finally, analysis of the 2 × 202/141(T) virus escape mutants isolated from the serum of female AG129 mice (Fig. 3i), as well as analysis of the barcode sequences of the viruses deposited into FRS of CD-1 mice (Fig. 6), demonstrated that sexually transmitted ZIKV genomes could be traced only to viruses that were found in the epididymis of a mating male partner, but not in any other MRS organs. These findings are in concordance with earlier reports that suggested a temporal correlation between ZIKV infection of the epididymal epithelia and shedding of ZIKV into mouse semen[19,21,26]. Together, our results demonstrate that the ZIKV that was sexually transmitted between 7 and 11 dpi was primarily produced by the cells of epididymal epithelium. These types of cells may secrete Zika virions into the lumen of epididymal tubules in a cell-free form[21], and these virions, in turn, are flushed out of the epididymis during the emission phase of the ejaculation (see Fig. 5c for schematics) and eventually expelled into the ejaculate and become sexually transmitted.

An important limitation of our immunodeficient AG129 mouse model of ZIKV infection is that it may not accurately recapitulate ZIKV pathogenesis in the MRS of humans or non-human primates (NHP) (reviewed in ref. [12]). It is possible that the aberrant type I and II IFN responses in this model may render the epididymal epithelium more susceptible to ZIKV, leading to an overestimation of the contribution of these cell types to sexual transmission of ZIKV. Therefore, more studies using an immunocompetent mouse model[54,55] or NHPs are needed to confirm the role of the epididymis in ZIKV shedding into semen during an acute infection.

**Testicular ZIKV infection does not contribute to the early (prior to 12 dpi) shedding of ZIKV into the semen and ST of the virus**. Between 7 and 17 dpi, natural ZIKV isolates replicate efficiently in the testis of both the immunodeficient and immunocompetent mice[17–20,22,26,55]. The genetically modified ZIKV clones used in this study (2 × scr, 2 × 141(T)) also attained high viral titers in the testis (Fig. 1d). However, we did not detect any evidence implicating testis as a source of sexually transmitted ZIKV genomes. For instance: (i) reduced accumulation of 2 × 202

(T)stb virus in the testis was not associated with a corresponding reduction in the ST rate of the virus from male to female AG129 mice (Fig. 3b, e); (ii) while escape mutants of the 2 × 202(T) virus that lost both mir-202 targets replicated efficiently in the testis, these mutants were not transmitted to the female mating partners (only viruses with intact miRNA targets were detected in the serum of these females) (Fig. 3d); (iii) barcode sequences of the ZIKV-lib/n11 viruses deposited into FRS of CD-1 mice at 10 dpi did not match sequences found in the testis of their male mating partners (Fig. 6).

Previously, using miRNA-targeted viruses, we showed that ZIKV disseminates from the testis into the epididymis of young (4–6 weeks old) AG129 mice by 12 dpi[22]. Here, using older (>10 week old) mice infected with ZIKV-lib/n11 viruses, we demonstrated that efficient ZIKV migration between these parts of the MRS occurs as early as 10 dpi (Fig. 5e), confirming that ZIKV can exit the testis with the efferent flow of sperm. However, barcodes of the ZIKV-lib/n11 viruses that were abundant in the testes (testis-specific barcodes) were not detected among viruses isolated from the vas deferens (Fig. 5e). This suggests that migration of testis-specific ZIKV virions through epididymis is either blocked or requires a longer time to complete, attenuating the shedding of testis-specific ZIKV-lib/n11 virions into the semen prior to 12 dpi. This may be similar to other viruses. For example, seminal SIV has been shown to originate from multiple genital organs of NHPs and migrate through the testis–epididymis–vas deferens axis during a long-term chronic infection[51].

In mice, under normal physiological conditions, it could take up to 9 days for maturing spermatozoa to pass through the entire epididymis[56]. Assuming that a secretion of ZIKV virions from the testis into epididymis begins at ~9 dpi (see virus 2 × 141(T)stb in Fig. 1d) and that migration rates of sperm cells and ZIKV virions through epididymis might be comparable, then the testis-produced ZIKV virions should not reach vas deferens prior to 18 dpi. Since the semen shedding and ST experiments in this study were terminated at 11 and 17 dpi, respectively, it is possible that there was not enough time for the testis-specific ZIKV virions to complete the passage through the epididymis to be emitted into the vas deferens. Alternatively, ZIKV infection may reduce the serum testosterone levels in mice[18,20,46], which would enhance spermatozoa migration through the epididymis[57,58], shortening the time for the testis-produced ZIKV virions to be expulsed with the ejaculate. It also remains to be investigated whether the ZIKV-induced epididymal inflammation and infiltration of leukocytes into the lumen of epididymal tubules[18,20,21] may affect the course of trans-epididymal migration of the testis-derived ZIKV virions.

Future studies are needed to elucidate the fate of the testis-derived ZIKV virions during their transition through the epididymis and the rest of the MRS, and whether these virions can be transmitted to the female partner after 11–17 dpi. However, unrestricted replication of the natural isolates of ZIKV[19,59–61] and ZIKV clones that were utilized in this study (see Fig. 6 and ref. [22]) in the CNS of AG129 mice results in a considerable mouse mortality after 14 dpi. This prevented us from analyzing the ZIKV pathogenesis in the MRS during the late phase of viral infection. Substantial alterations to experimental approaches would need to be made to increase mouse survival without affecting viral replication in the MRS and sexual behavior of infected animals. One possibility for reduction of ZIKV neurovirulence in AG129 mice would be to incorporate into ZIKV genome additional miRNA targets that are selectively expressed in the mouse CNS, but not in the MRS[22,62]. It would also be interesting to evaluate ZIKV pathogenesis in the MRS of an immunocompetent host (e.g., NHPs and immunocompetent

**ZIKV infection of the MRS accessory glands may play a supplementary role in the ST of the virus**. Natural ZIKV isolates can replicate in the SV/P of mice and non-human primates[23,25]. However, our study suggests that these accessory glands of the MRS mostly play a supplementary role in the ST of the virus. They were able to support a relatively low-level (compared to the testis and epididymis) replication of our model ZIKV ($2 \times$ scr), mainly in the epithelial cells. As we anticipated, targeting of ZIKV genome for epithelial miRNA mir-141 effectively restricted ZIKV replication in the SV/P, confirming the epithelial tropism of ZIKV. However, these accessory glands, in addition to the epididymis, may also contribute ZIKV output to the ejaculate, albeit more variably. One limitation of our experiment with the barcoded ZIKV is that we cannot exclude the possibility of contamination of the SV/P samples by the content of the adjacent vas deferens ampulla where the infected sperm could be stored downstream of the epididymis. These findings, together with the notion that the vasectomy attenuates shedding of infectious ZIKV virions and non-infectious viral RNA into seminal fluids of mice[19], suggest that MRS accessory glands may play only supplementary role in the ST of the virus under normal physiological conditions. Nevertheless, it is important to stress that this does not diminish the risk of ST of ZIKV in the vasectomized cases, which can potentially occur outside the acute phase of ZIKV infection[34].

**Abolishment of the epithelial tropism of ZIKV reinforces the safety of a live attenuated vaccine candidate**. A long-term persistence of ZIKV in the MRS and its propensity for horizontal transmission warrant the effort to eliminate these attributes in a live-attenuated ZIKV vaccine. Here, we demonstrate that ZIKV genome targeting for mir-141 restricts virus infection of the epithelial cells in the epididymis, seminal vesicles, and prostate. It also directly prevents ZIKV shedding into the semen, blocks ST, and attenuates to some extent ZIKV replication in the female reproductive system (Figs. 1–4, S7). Targets for this miRNA constitute an important attenuating component of our previously characterized ZIKV vaccine candidate C/3′NCR-mir(T) virus[22]. Not surprisingly, male AG129 mice infected with this virus were not capable of transmitting the virus to the mating females, and the infectious virus was not detected in the epididymis and SV/P of male mice at 17 dpi (Fig. 7d, f). As discussed above, it remains unclear whether the natural isolates of ZIKV produced in the testis could be sexually transmitted by infected males after 17 dpi. However, targets for the testis-expressed mir-202 incorporated into the genome of the C/3′NCR-mir(T) vaccine candidate virus appear to be sufficient to completely abolish the testicular tropism of the virus (Fig. 7c)[22], ensuring that transmissibility of vaccine candidate virus(es) containing targets for miRNAs mir-202 and mir-141 through sexual contact will remain completely restricted.

## Methods
**Statement of compliance**. All experimental protocols were approved by the NIH Institutional Biosafety Committee. All animal study protocols were approved by the NIAID/NIH Institutional Animal Care and Use Committee (IACUC) and performed in compliance with the guidelines of the NIAID/NIH IACUC. The NIAID DIR Animal Care and Use Program acknowledges and accepts responsibility for the care and use of animals involved in activities covered by the NIH IRP's PHS Assurance D16-00602, last approved 6/10/2019.

**Plasmids and viruses**. Construction of infectious cDNA clones encoding ZIKV-NS3m, $2 \times$ scr, $2 \times 202$(T), $2 \times 141$(T), $2 \times 202/141$(T), C/3′NCR-mir(T), C/3′NCR-scr, and 3′NCRΔ20 viruses has been reported previously[22,45]. Complete sequences of these plasmids are available from the authors upon request. To construct ZIKV-

lib/n11 plasmid library, we amplified 846 bp DNA fragment (Fig. 5a; BssHII-NsiI fragment) corresponding to the 3′ terminal part of NS5 gene and 5′ end of the 3′ NCR of $2 \times$ scr virus using Phusion® High-Fidelity DNA Polymerase (New England Biolabs [NEB], MA), a primer pair ZV-9574-F and Library-R [Table S2, (Integrated DNA Technologies, IA)], and $2 \times$ scr plasmid as a template. Degenerated primer Library-R contains a stretch of 11 randomized nucleotides, which partially substitutes scr sequences located at the 5′ terminus of the 3′NCR of the $2 \times$ scr virus (Fig. 5a). The amplicon was digested using BssHII and NsiI endonucleases (NEB, MA), followed by 1% agarose gel electrophoresis and recovery of 527 bp DNA fragment using Zymoclean™ Gel DNA recovery kit (Zymo research, CA). This fragment was cloned into $2 \times$ scr plasmid using unique sites for BssHII and NsiI endonucleases, followed by transformation of E. coli cells (strain MC1061) with the ligation reaction. Colonies of E. coli ($n \sim 93{,}000$) containing recombinant DNA were propagated for 18 h at 37 °C in 30 agar plates (Falcone, $100 \times 15$-mm style) containing 50 μg/mL of ampicillin. To validate cloning efficiency, cells from eight randomly selected colonies of E. coli were amplified, and plasmid DNAs were analyzed by the sequencing of the 3′NCR of ZIKV cDNA (Fig. S8). Each agar plate containing E. coli colonies was washed twice with 5 mL of LB broth supplemented with 50 μg/mL ampicillin. Washing broth from all plates was combined into single flask, incubated in bacteriological shaker for 3 h at 37 °C at 200 rpm, followed by plasmid extraction using Endo Free Plasmid Maxi kit (Qiagen, Germany). Plasmid integrity was verified by restriction endonuclease digestion and Sanger sequencing.

**Recovery of viruses from infectious cDNA clones**. Vero cells (Cercopithecus aethiops kidney) were maintained at 37 °C and 5% $CO_2$ in Opti-Pro medium (Invitrogen, CA) supplemented with 4 mM L-glutamine[63]. Recovery of ZIKV-NS3m, $2 \times$ scr, $2 \times 202$(T), $2 \times 141$(T), $2 \times 202/141$(T), C/3′NCR-mir(T), C/3′NCR-scr, and 3′NCRΔ20 viruses was performed using plasmid DNA trasfection method[45]. Briefly, for each infectious clone 2.5 μg of plasmid DNA was transfected into $1.5 \times 10^6$ Vero cells seeded into one 12.5-cm² flask using Lipofectamine 2000 transfection reagent (Invitrogen, CA). To achieve high 'barcode' heterogenicity, ZIKV-lib/n11 plasmid library was transfected into four 25-cm² flasks of Vero cells using Lipofectamine 3000 reagent (Invitrogen, CA), which provides superior efficiency of plasmid DNA transfection as compared to Lipofectamine 2000 reagent. For each flask, we seeded $3 \times 10^6$ Vero cells in DMEM supplemented with 10% FBS (HyClone laboratories, UT) and 1× penicillin–streptomycin–glutamine solution (Invitrogen, CA). Next morning DMEM was replaced with Opti MEM (Invitrogen, CA), and cells in each flask were transfected with 8 μg of ZIKV-lib/n11 plasmid DNA according to manufacturer's instructions. At day 3 post transfection cell culture supernatants from all four of these flasks were combined, clarified by 5 min centrifugation at $3000 \times g$, supplemented with 1× SPG solution (218 mM sucrose, 6 mM L-glutamic acid, 3.8 mM $KH_2PO_4$, 7.2 mM $K_2HPO_4$, pH 7.2)[64], aliquoted, and stored at −80 °C. Titer of recovered viruses was determined by plaque assay in Vero cells[45]. Viruses recovered after transfection of recombinant ZIKV clones into Vero cells were used in all animal experiments without additional propagation.

**Mouse strains**. A colony of AG129 mice was maintained at the NIAID/NIH animal facility at 20.6–23.9 °C, 30–70% relative humidity under a 14 h light and 10 h dark photoperiod. The colony was originally established from a breeding pair purchased from Marshall BioResources. Female CD-1 were purchased from Taconic Farms.

**Growth kinetics of recombinant ZIKVs in the tissue organs of adult male AG129 mice**. L-15 media (Invitrogen, CA) supplemented with 1× SPG solution (L-15/1×SPG) was used to dilute ZIKV to a concentration of $10^7$ pfu/mL, and male AG129 mice (10–20-week-old) were inoculated IP with 0.1 mL of each virus (dose: $10^6$ pfu/mouse). High virus dose and IP route of inoculation were chosen to compensate for moderate attenuating effect associated with insertion of heterologous sequences (such as miRNA targets) into 3′NCR of ZIKV genome[22,65]. Mice were bled at 1 dpi and euthanized at 6, 9, 12, and 17 dpi. To minimize total number of aminals used in this study, organs of MRS for time point 17 dpi were dissected from male mice that participated in the ST sudies with non-infected AG129 female mice (see below). Mouse serum and MRS organs (a pair of testes, pair of epididymis, pair of vas deferens, and combined seminal vesicles and prostate (SV/P) specimen) were harvested and stored at −80 °C. The SV/P specimen consisted of a pair of seminal vesicles, an anterior prostate and a ventral prostate (Fig. S3). ZIKV titer in mouse tissues was determined by plaque assays in Vero cells using duplicate wells of 24-well plates as was described previously[22,63]. Light weight of vas deferens and significant variations in the amount of fat tissue collected during dissection of epididymis compelled us to perform normalization of ZIKV titer not to a pfu per gram of tissue, but rather to pfu per whole organ. For that epididymis and vas deferens were not weighted but homogenized in 1 mL of L-15/1×SPG solution. In contrast, testes and SV/P were weighed and triturated in 9 volumes of L-15/1×SPG solution (making 10% organ homogenate). Viral titers in these organs were normalized to pfu per gram or tissue. Serum was diluted in 5 volumes of L-15/1×SPG, followed by Vero cells titration.

At 5 dpi, virus-infected Vero cell monolayers were observed under light microscope for ZIKV-induced cytopathic effect (CPE). If the titrated sample induced CPE (plaque formation), the overlay medium (Opti-MEM containing 1%

methylcellulose (Invitrogen), 2% heat-inactivated FBS, 4 mM L-glutamine) was aspirated from duplicate wells of Vero cells infected with lowest dilution of titrated specimen (containing highest number of ZIKV plaques). Cells were carefully washed with 0.25 mL of Opti-Pro. Washing medium from duplicate wells was combined together and 0.14 mL of it was used for ZIKV RNA extraction using the QIAamp Viral RNA Mini kit (Qiagen). Subsequently, Vero cells in all wells of the 24-well plate were fixed with 100% methanol and stained with crystal violet.

**ST of ZIKV from male to female AG129 mouse.** Male AG129 mice (10–20 weeks-old) were infected IP with $10^6$ pfu of ZIKVs. At 7 dpi, each infected male was placed into separate cage containing a single non-infected AG129 female mouse. Mice were allowed to mate for 10 days. At 3, 5, and 10 days of mating females were bled, and virus titer in the serum was determined by the plaque assay in Vero cells. At 17 dpi, male mice were sacrificed, and MRS tissues were collected and stored at −80 °C. Female mice were returned to cages and monitored for signs of neurological disease for 18 days (28 days post mating). Brains from female mice that succumbed to paralysis were collected, and ZIKV titers in the brain homogenates were determined in Vero cells. Blood from the remaining (healthy) female mice was collected at 28 days post mating, and a titer of ZIKV-specific NA in the serum was determined using PRNT$_{50}$ assay described previously[66]. The female was considered infected if: (i) ZIKV-specific NA were detected in the female serum and/or (ii) infectious ZIKV was detected in the serum or brain of the female by plaque assay in Vero cells.

The titer of 2 × scr, 2 × 202(T), 2 × 141(T), and 2 × 202/141(T) viruses in the organs of MRS was determined only for those male mice that transmitted the virus to the female mating partner, and only if infectious virus was also detected in the serum and/or the brain of this female by plaque assay in Vero cells. If serially diluted (titrated) male or female sample (organs or serum) induced CPE in Vero cells seeded in 24-wells plate, at 5 dpi viral RNA was extracted from titration plate from the Vero cells, which were infected with the lowest dilution of titrated specimen (contains highest number of ZIKV plaques) as described above. The titer of the C/3′NCR-mir(T) in the organs of MRS at 17 dpi was determined for all male mice (n = 15) participating in the mating experiment, while the titer of C/3′NCR-scr and 3′NCRΔ20 viruses in the MRS at 17 dpi was determined only for 10 and 5 randomly selected male mice, respectively.

**Replication of recombinant ZIKVs in female AG129 mice following intravaginal exposure.** Intravaginal infection of 10–20 week-old AG129 mice with ZIKVs was performed according to previously described protocols[67]. It was shown that mouse FRS is the most susceptible to ZIKV infection during progesterone-high diestrus phase of estrous cycle[68]. To ensure synchronization of estrous cycle at diestrus phase, all females were subcutaneously injected with 0.1 mL (400 mg/mL) of Depo-provera 7 days prior to virus exposure (Pfizer, NY). Immediately prior to inoculation, mucus from mouse vaginal cavity was removed with sterile cotton swab moistened with PBS, followed by injection of 10 μL of virus diluted L-15/1×SPG ($10^5$ pfu/mouse) into the vagina using a pipette tip. In the experiment-1, females (n = 10 per virus) were bled on 1, 3, 6, 9, and 13 dpi to determine virus titer in serum, and on 28 dpi to determine NA titer against ZIKV using PRNT$_{50}$ assay. In addition, brains from female mice that succumbed to paralysis prior to 28 dpi were collected, following determination of ZIKV titers by plaque assay in Vero cells. A female was considered infected with ZIKV via intravaginal route if ZIKV-specific NA were detected in the female serum at 28 dpi or if virus was detected in the brain of the morbid female. In the experiment-2, females were sacrificed at 1, 3, 6, and 9 dpi (n = 5 mice for each time interval), followed by vagina, cervix, and uterus dissection.

**Studies of ZIKV shedding into the semen of AG129 mice.** Semen shedding experiments were conducted using CD-1 strain of female mice. In contrast to AG129 strain, CD-1 mice have intact IFN-signaling pathway and are resistant to ZIKV infection[69]. Inability of CD-1 to propagate ZIKV in their FRS makes them a better model for assessing the amount of ZIKV in the semen deposited into FRS during coitus.

In the experiment-1, 10–20 weeks old male AG129 mice were infected IP with $10^6$ pfu of 2 × scr or 2 × 141(T) virus (n = 20 mice/virus). At 7 dpi each infected male was placed into separate cage containing two non-infected female CD-1 mice. Mice were allowed to mate for 4 days. Females were observed twice a day (morning and afternoon) for presence of vaginal (copulatory) plug. Mice that developed vaginal plugs were sacrificed, and FRS were dissected and triturated in L-15/1×SPG. Viral load in the FRS was determined by plaque assay in Vero cells. To decrease limit of virus detection (increase sensitivity of virus detection assay) 0.5 mL of 10% FRS homogenate was used to infect monolayer of Vero cells in 25-cm$^2$ flask. At 5 dpi cells were analyzed under light microscope to determine whether virus-induced CPE had developed, followed by viral RNA extraction from 0.14 mL of Vero cell supernatant for RT-PCR and Sanger sequencing analysis. If male mouse deposited semen to both CD-1 mice housed in the same cage, the titer of ZIKV in the semen of this male was reported as mean ZIKV titers in the FRS of both females participating in mating.

In the experiment-2, male AG129 mice were infected IP with $10^6$ pfu of ZIKV-lib/n11 viruses and mouse serum was collected at 1 dpi. Male mice were allowed to mate with two non-infected female CD-1 mice from 7 to 11 dpi. The male mouse was sacrificed immediately after the vaginal plug was detected in one of the two CD-1 mating partners, followed by dissection of the MRS organs and the brains from the male, and FRS from the female mouse. Additionally, MRS and brains were collected from three males who did not deposit semen to female partner by 11 dpi. Organs and serum specimens were homogenized in or diluted with L-15/1×SPG solution as described above followed by plaque assay. To minimize the effect of Vero cell passaging on ZIKV-lib/n11 virus diversity, viral RNA was extracted directly from 0.14 mL of male and female organ homogenates or 5× diluted serum sample (and not from the ZIKV-infected Vero cells) using the QIAamp Viral RNA Mini kit (Qiagen).

**Detection of ZIKV immunoreactivity in the MRS of mice.** Male AG129 mice were mock inoculated (n = 1) or infected IP with $10^6$ pfu of 2 × scr or 2 × 141(T) virus (n = 4 mice/virus). At 9 dpi, mice were euthanized, and MRS tissues were fixed in 4% paraformaldehyde. Pair of epididymis, pair of vas deferens, pair of seminal vesicle, and prostate (ventral) from each mouse were embedded in histological grade paraffin. Blocks were sectioned at ~5 μm and thereafter stained with hematoxylin–eosin (H&E) for examination by light microscopy. Immunohistochemical (IHC) staining was performed on a Leica Bond-RX automated system according to manufacturer recommended protocol. Tissue sections were heated to 72 °C for 30 min in Bond Dewax Solution (Leica) then rehydrated with absolute alcohol washes and 1× ImmunoWash (ACR-024, StatLab). After that sections were heated to 100 °C for 20 min in Bond Epitope Retrieval Solution 1 (Leica) for heat-induced epitope retrieval (HIER). After exposure to peroxide block (Leica) for 5 min, tissues were incubated with the anti-ZIKV NS2B antibody (1:500 dilution factor; GTX133308, GeneTex). The NS2B is a non-structural protein that can be detected in the host cell during active virus replication. The tissues were then incubated with secondary antibody (anti-rabbit Poly-HRP-IgG, DS9800, Leica), rinsed with 1× ImmunoWash and viral antigen was detected using 3,3′diaminobenzidine (DAB) chromogen using the Bond Polymer Refine Detection Kit (DS9800, Leica). Sections were then counterstained with hematoxylin. ScanScope AT2 was used to acquire digital images of whole tissue section at ×40 magnification. Digital slides were analyzed using Aperio Spectrum eSlide Manager and ImageScope software (Version 12.4).

**Detection of ZIKV-specific NA in mouse serum.** The titer of ZIKV-specific NA in mouse serum was determined by PRNT$_{50}$ assay as described previously[22,66]. The serum was considered positive for ZIKV-specific NA if 1:10 dilution of serum caused >50% reduction in a number of ZIKV-NS3m plaques in Vero cells.

**Stability of miRNA targets and scr sequences in viruses isolated from mouse organs and serum.** A pair of ZIKV-specific primers ZV-10044-F and ZV-10722-R (Table S2) was used to amplify a region containing miRNA target (or scr) sequences inserted into the 3′NCR of ZIKV. Viral RNA was extracted from supernatants of Vero cells, which were infected with a virus in the serum or homogenate of mouse organs (see above). The RT-PCR reaction (35 cycles) was carried our using the Transcriptor One-Step RT-PCR Kit (Roche). Amplicons were purified by 1% agarose gel electrophoreses and sequenced by Sanger method using ZV-10722-R and/or ZV-10044-F primers, BigDye™ Terminator v3.1 (Applied Biosystems) and 3730 DNA Analyzer (Applied Biosystems).

**Deep-sequencing analysis of ZIKV-lib/n11 virus.** To reduce/remove residual amount of plasmid DNA, the sample containing inoculum for mouse infection was treated with micrococcal nuclease (NEB, MA) for 2 h at 37 °C, followed by viral RNA extraction. Virus RNA from all other samples was extracted without micrococcal nuclease treatment (see above). The region in the ZIKV-lib/n11 virus genome containing barcode sequences was amplified for 35 cycles using the Transcriptor One-Step RT-PCR Kit (Roche) and ZV-10722-R/ZV-10044-F primer pair, followed by additional 20 PCR cycles using inner pair of primers ZV-10239-F and ZV-10489R and LongAmp Taq 2× Master Mix (NEB, MA). RT-PCR products were purified using Agencourt AMPure XP Reagent (Beckman Coulter). The purified DNA was eluted in 40 μl of DEPC-treated water. DNA concentration was measured with Qubit 2.0 fluorimeter (Invitrogen, Life Technologies), and the DNA frozen at −20 °C until further use. Illumina libraries were prepared using NEBNext Ultra II DNA Library Prep Kit (NEB, MA) and NEBNext Multiplex Oligos for Illumina, Sets 1 and 2 (NEB, MA). Library quality was assessed using 4200 TapeStation Bioanalyzer Instrument (Agilent Technologies) with High Sensitivity D1000 Reagents (Agilent Technologies) and High Sensitivity D1000 ScreenTape (Agilent Technologies). Deep sequencing was done using MiSeq instrument (Illumina) with MiSeq Reagent Kit v2, 500-cycles (Illumina).

Bioinformatic analysis was performed using in-house SWARM software (Supplementary Software 1). Sequence reads were sorted by unique molecular identifiers (barcodes) introduced into the viral genome (see above). Since point nucleotide substitutions may have been introduced during sample processing and sequencing, barcodes differing by only one nucleotide were combined into a single group. To identify the source of virus in each tissue, respective samples were compared to each other by barcode profiles (quantitative distribution of unique barcodes (frequency), which is characteristic for each sample). To determine

number of barcodes in reads[90] for each organ we counted minimal number of barcodes in the sample, frequencies of which together would be equal to or greater than 90% of all identified barcodes. The 90% cutoff was selected to eliminate non-replicating genomes (background), which could have been seeded in the organ during initial ZIKV-lib/n11 dissemination. In addition, it eliminates from consideration most barcodes that differ from the most abundant barcodes in the sample by two or three nucleotides. Genetic relatedness of such barcodes to dominate barcode sequences (see Supplementary Data 1 for an example of barcode distributions in the organs collected from mice in mating pair # 1 in Fig. 6) is a clear indication that they represent an artifact of sample processing. To calculate a combined frequency of common barcodes, we identified all barcodes which are simultaneously present among reads[90] barcodes in both organs in question. Individual frequencies of these barcodes were combined, and a resulting value was divided by two for normalization.

**Reporting summary**. Further information on research design is available in the Nature Research Reporting Summary linked to this article.

## Data availability

The authors declare that all data supporting the findings of this study are available within the paper and its Supplementary Information. Full-length sequences for all viruses and for all cDNA infectious clones used in this study are available from the corresponding author upon request. Microarray data used for construction of graphs in Fig. 1a, 1b were obtained from https://static-content.springer.com/esm/art%3A10.1038%2Fsdata.2014.5/MediaObjects/41597_2014_BFsdata20145_MOESM75_ESM.zip[40]. Source data are provided with this paper.

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

## Acknowledgements
We thank Evgeniya Volkova and Dr. Charles E. McGee for critically reviewing the manuscript. This work was supported by the Division of Intramural Research Program of the National Institute of Allergy and Infectious Diseases, National Institutes of Health.

## Author contributions
A.G.P., K.C., and K.A.T. designed the experiments; A.G.P., O.A.M., G.L., H.K., B.M.N., T.Z., I.M., K.C., and K.A.T. performed the research; A.G.P., O.A.M., I.M., K.C., and K.A.T. analyzed the data; A.G.P., O.A.M., I.M., K.C., and K.A.T. wrote the paper. All authors reviewed the final draft of the manuscript.

## Funding

## Competing interests
The authors declare no competing interests.
