## [Peer Review File · Nature Communications]

Reviewers' Comments:

Reviewer #1:

Remarks to the Author:

By using highly innovative approach and well-designed experiments, the authors have delineated the relative contribution of the components of the mouse male reproductive system (MRS) on male-to-female sexual transmission of the ZIKV. The data is well represented and suggests that during early infection in the AG129 model, seminal shedding of the virus originates from infected epididymal tissue. There are no major concerns regarding the data or the approach, however applicability of this data to understand sequence of events in humans is a matter of serious concern due to the choice of the animal model. Due to major limitations of the model, conclusions may not reflect what occurs during ZIKV infection in human males.

AG129 mice are a poor model to evaluate ZIKV tissue tropism and/or trafficking of disseminated virus. IFN-I response is an important determinant of tissue and cell tropism and IFN-II response is important for regulating immune cell trafficking and activation in response to virus infection. In this model that lacks both these important signaling, the virus is not cleared by most of the tissues and exhibits severe inflammation in the testes- an event not reported in ZIKV-infected humans. Thus, there are inherent limitations to using AG129 model for the study of persistent ZIKV infection in the MRT. Therefore, after initial studies using this mouse model, many groups have moved to other slightly better animal models for ZIKV studies including use of anti-IFNa/b antibody just before infection.

Thus, its cannot be ruled out that the data reported in this study, specially that the epithelial cells of the epididymis (rather than the cells of the testis) are the source of the sexually transmitted ZIKV genomes is a feature specific to this animal model only. It is highly likely that these impaired immune responses in this mouse model may alter tissue tropism and tissue homeostasis to alter trafficking of the virus.

This reviewer believes that at least some of the key data from this study should be validated using other mouse model otherwise it might set an incorrect precedent/explanation for ZIKV transmission via sexual route. Further, the weakness of this model and the results should be clearly explained.

Other comments:

Line 122 – authors note that 2x141(T)mut was not detected in testis, but do not show any data points in Fig 1D for this virus or do they state data not shown. And also vice versa for 2x202(T)mut in epididymis (Fig 1E).

Male mice were infected IP with high titers of virus, this is not a natural route of infection.

Reviewer #2:

Remarks to the Author:

Using two original and elegant approaches - miRNA-targeted ZIKV clones to block ZIKV replication in various parts of the male genital tract, and a library of ZIKV genomes with unique molecular identifiers to trace individual ZIKV genomes during male to female sexual transmission - Pletnev et al. define the main source of ZIKV virions transmitted to female sexual mice partner during the early course of infection of immunocompromised AG129 mice. Their analyses are clear and meticulous. They pinpoint epithelial cells of the epididymis, rather than cells of the testis, prostate and seminal vesicles as a leading contributor to the sexually transmitted ZIKV in this model. They also eliminate any potential for sexual transmission of a previously developed live-attenuated ZIKV vaccine candidate C/3'NCR-mir(T) containing multiple targets for miRNAs, including targets for testicular and epithelial cells specific miRNA.

Pletnev et al. provide a well-designed study of the cellular and molecular mechanisms underlying ZIKV

male-to-female sexual transmission a mouse model. However, the use of immunocompromised mice is a significant limitation of the study. As a main concern, an overall consideration of the limits of this model to decipher the sexual transmission of ZIKV in human and non-human primate models is missing. Moreover, some figures' interpretations should be clarified.

Major comments:

- The strong deleterious effect of ZIKV on the male reproductive tract of immunocompromised mice such as AG129 drastically differs from that in ZIKV-infected men and NHP (both ex vivo and in vivo), in whom organ morphology is generally preserved despite the infection and semen parameters only transiently and mildly impacted. This needs to be clearly stated in the manuscript. Thus, in immunocompetent hosts such as human or NHP, a number of cell types within the MRT may not be readily permissive to ZIKV infection. This might lead to important differences in tropism and hence in the contributions of male genital cells/organs to ZIKV shedding into semen. The mammalian testis is an immunoprivileged organ with immunosuppressive properties as well as weakened IFN responses (see Le Tortorec et al 2020 Physiological reviews). In those preserved immunological conditions, the testis might represent a more favourable environment for ZIKV replication and persistence, and hence play a more prevalent role than in this immunocompromised animal model. Conversely, a functional IFN system may considerably reduce the infectivity of the epididymis and in turn decrease its relative contribution to ZIKV seminal shedding in semen. Therefore, the use of an immunocompromised model such as AG129 mice can severely bias the relative contribution of male genital organs to ZIKV shedding into the semen, and this needs to be properly acknowledged as an important limitation for extending those conclusions to humans. The authors should therefore discuss the limits of their model regarding the prediction of trafficking and persistence of ZIKV in human and NHP male genital tract in the light of these elements. For the same reasons, the title of the article and abstract should specify that this work was performed in immunocompromised mice that were acutely infected.

- Figure 1E and 1F, lines 163-169 : the authors state that the absence of apparition of escape mutation of 2x202 in the epididymis and vas deferens over time indicate that the excurrent transport of ZIKV from testis is quite inefficient. Bearing in mind the timing for spermatozoa excretion from testis into semen and that escape variants are detected in the testis only from 12 dpi, they cannot exclude at this stage the requirement for longer kinetics to seed infection in these organs.

- Another important point is that the seminal shedding examined in this work only corresponds to acute infection, and not persistence over several weeks or months as reported in men up to over one year. As expected and mentioned by the authors, mating occurred in the first days of contact between female and 7 day infected males, so at best ejaculates reflect d12-infected mice. While this issue is acknowledged in the discussion, it needs to be clearly indicating from the title onwards. Knowing the physiology of the MRT, it is indeed unlikely that the testis would represent a major contributor to virus release in semen during the early stage of the infection, and the results of this work are therefore not surprising in that respect. The main contribution of the testis to semen is the production of sperm, which take about 35 days in the mouse (74 days in men) to form from spermatogonia, which may represent the primary initial targets of ZIKV in the seminiferous tubules. Testicular spermatozoa then transit through the epididymis for about 10 days in mammals, before being released in semen. Seminal fluid is at >90% constituted from the secretions of the prostate, seminal vesicles and epididymis. Thus one expect testis contribution to semen infection to be mostly through infected cell release, rather than through secretions. This could play a role in the long term rather than during the acute stage. Moreover, the results suggest that the timing of virus spillover across accessory glands could differ according to their respective anatomical distances.

- Line 81, information about the physiological role of mir-202 in the seminiferous tubules should be added, similarly to mir-141, and expression in testis interstitial cell discussed. Moreover, the authors should comment on how the miR-engineered virus may affect organ functions such as epithelial cell barrier integrity in the epididymis and spermatogenesis in the testis (e.g by "sponging" the miRNA from their physiological targets)?

- How do the authors explain that miR-202 abrogate replication in the testis, although mir202 does not target macrophages, an important infected cell type in the testis?

- Fig3 E-H: this figure should show the level of mir 141 in the testis and other organs to make sure that lack of transmission did not reflect absence of MRT infection by the variant in this experiment.

Why is this data missing? The indirect measurement of ZIKV mir-141 in semen through virus load measure in female mice only confirms the lack of transmission of this virus in acute stage but does not demonstrate MRT infection (eg testis) in those mice.

- Figure 5E: even if no common barcodes were detected in between testis and vas deferens, it could be informative to analyse the presence of common barcodes in between the testis and the SV/P.

- Fig 6 and lines 386-394: it is striking that the majority of the strains detected in the ejaculate arise from the vas deferens, suggesting that viruses are flushed out from epididymis tail/vas deferens during ejaculation, where sperm are stored. Can the authors assess whether the viruses detected in the ejaculate are bound to sperm ?

Specifics comments:

- Line 41: the choice of references is questionable as there are more comprehensive/recent reviews and articles to quote.

- Line 57-60: In Osuna et al, persistent (28 dpi) infected cell foci were reported in testis, prostate and seminal vesicles. Therefore it is not correct to state that only testis and epididymis were reported in NHP as being persistently infected. Persistence of ZIKV RNA was also detected in vasectomized men, ruling out epididymis and testis as a source in these cases. Furthermore, the level of MRT infection across NHP studies is overall very low, with discrepancies as to the target organs and little seminal shedding. Thus caution must be exerted and no definite statements should be made from this model.

- Lines 60-61: the authors may reinforce the message of a persistent infection in the testis and/or epididymis with references to studies that demonstrate the long term shedding of ZIKV positive spermatozoa in human semen.

- Figure 1E, D: The authors should comment on the fact that levels of mir-202 stb virus in the testis at day 6 are similar to that of other viruses (and decrease thereafter), whereas mir-141 stb virus in the epididymis at day6 is several log lower than others, but increases over time. The explanation that the increase of mir-141 in the epididymis is due to transport of virus from the testis could appear challenged to the reader by the fact that this is not observed for ZIKV-NS3m and scr, which also replicate in the testis and show even more increase replication in testis than mir-141 over time. Differences in viral load threshold likely account for this but this needs to be explained. To consolidate the idea that excurrent transport plays a role, the authors could refer to their previous published data showing that at 12 dpi ZIKV positive antigen immunoreactivity for mir-141 stb was exclusively found in luminal cells. Overall, this part of the ms is a bit confusing and hard to follow.

- The authors suggest that epithelial cells are the main infected cell types in the epididymis and seminal vesicles/prostate. What then explains the much lower infection levels of seminal vesicles/prostate versus epididymis, especially if those organs are additionally seeded with virus coming from the epididymis?

- Figure 3C, the measure of infectious titer in female serum would be more convincing to demonstrate the equivalent virus infectivity properties of the viral clones.

- Line 238, there is a mistake in the figure reference, 3E instead of 3B.

- Line 303, edition mistake after the references.

- Lines 325-327: a reference that could be added, in support of the authors' hypothesis on the existence of tissue-specific viral variants in the MRT, is one regarding SIV organ-specific sequences/phylogeny in the MRT of macaques (Houzet et al, J Virol 2018), which is also showing virus intermingling between testis and epididymis. This could be commented on in the discussion in order to extend the authors' observations on ZIKV dissemination in the MRT to other viruses and models.

- Lines 364-365: it would have been interesting to test further time points using 17 day-infected male mice to determine whether testis-borne viral strains travel out of the epididymis, with or without ejaculation.

- Lines 449 and 471: please precise "during the acute stage" for epididymis as a main contributor of sexually transmitted ZIKV in immunocompromised mice.

- Line 524, there is a misspelling for the word "stability"

- Line 528, there is a misspelling for the word "transmitted"

- Line 538: the immunocompetent mouse model developed by Robinson et al (Nature communication, 2018) could also be of interest for further MRT studies.

- Line 485: an important point to discuss here is how ZIKV target cells in the testis compare between immunocompromised and immunocompetent mice models (Robinson et al, Nature Communications 2018) and human testis (Matusali et al, JCI 2018).
- Line 560: mention here sexual transmission by a vasectomized man outside acute stage (Arsuaga et al, Lancet Infect Dis 2016)
- Line 635, there is a misspelling for the word "strain"
- Line 709, there is a misspelling for the word "contrast"
- Line 753, "primary" should be "secondary".

Reviewer #3:

Remarks to the Author:

The paper by Pletnev et al. analyzed the impact of the different cell types of the mouse male reproductive system on Zika virus sexual transmission. By using ZIKV clones engineered with different micro-RNA target sequences to specifically decrease infectivity in different organs and ZIKV libraries based on unique barcodes, they suggested that epithelial cells of the epididymis are the main actors involved in sexual transmission. Then, they highlight that their findings could be useful for the design of safer live attenuated ZIKV vaccine candidates. Overall the work is well conceived and innovative. Moreover, the authors have a track record in the current topic. The statistical analysis are OK. However, I suggest few experiments to better support the conclusions drawn.

Major concerns:

To further support the role of the epididymal epithelium as a source of the ZIKV genomes/virions secreted into the semen:

- i) Is there any possibility to only dissect the target infected tissue and sort these cells by flow cytometry using a virus antibody or targeting a GFP-like viral construct?
- ii) Use this tissue to extract viral RNA and perform negative-strand-specific RT-PCR to confirm viral replication. I would do the same on the other tissues/organs of the MRS
- iii) More experiments to support this key claim could be address on DC2 cells (immortalized mouse distal caput epididymal epithelial cell line).

Minor comments:

What about the mir-141 in the female mice?

Response to the reviewers

We thank the reviewers for their insightful and helpful comments and recommendations. Please note that line's references correspond to the manuscript with accepted track changes.

Answers to reviewers' comments.

Reviewer #1 (Remarks to the Author):

There are no major concerns regarding the data or the approach, however applicability of this data to understand sequence of events in humans is a matter of serious concern due to the choice of the animal model. Due to major limitations of the model, conclusions may not reflect what occurs during ZIKV infection in human males.

AG129 mice are a poor model to evaluate ZIKV tissue tropism and/or trafficking of disseminated virus. IFN-I response is an important determinant of tissue and cell tropism and IFN-II response is important for regulating immune cell trafficking and activation in response to virus infection. In this model that lacks both these important signaling, the virus is not cleared by most of the tissues and exhibits severe inflammation in the testes- an event not reported in ZIKV-infected humans. Thus, there are inherent limitations to using AG129 model for the study of persistent ZIKV infection in the MRT. Therefore, after initial studies using this mouse model, many groups have moved to other slightly better animal models for ZIKV studies including use of anti-IFNa/b antibody just before infection.

Thus, its cannot be ruled out that the data reported in this study, specially that the epithelial cells of the epididymis (rather than the cells of the testis) are the source of the sexually transmitted ZIKV genomes is a feature specific to this animal model only. It is highly likely that these impaired immune responses in this mouse model may alter tissue tropism and tissue homeostasis to alter trafficking of the virus. This reviewer believes that at least some of the key data from this study should be validated using other mouse model otherwise it might set an incorrect precedent/explanation for ZIKV transmission via sexual route. Further, the weakness of this model and the results should be clearly explained.

Response 1.1:

We agree with Reviewers #1 and #2 that the results obtained using immunodeficient mice model cannot be directly applied to infer outcome of viral infection in humans. To acknowledge this limitation, we modified both the title and the abstract to emphasize that presented results were obtained using immunodeficient model of Zika virus infection. We also added a new paragraph to Discussion describing limitations and the relevance of AG129 mouse model (Line492), which starts with "An important limitation of our immunocompromised..."

We also agree with both reviewers that additional studies using immunocompetent animals will be needed to validate conclusions of this investigation. However, there are several factors associated with unique aspects of ZIKV biology that restrain us from evaluating ZIKV infection events in the MRS of an immunocompetent host in this particular study:

A - Natural isolates of ZIKV cannot efficiently antagonize IFN responses in immunocompetent mice via NS5-mediated inhibition of STAT2 signaling (PMID: 27212660). This effectively eliminates all natural strains of mice as a satisfactory model for ZIKV pathogenesis. The exception/alternative to this rule might be hSTAT2 KI mice model for ZIKV infection (PMID: 29746837). This immunocompetent model recapitulates ZIKV infection of the placenta and trans-placental transfer of the virus to the fetus (PMID: 29746837). However, there are no published reports that indicate whether ZIKV can infect MRS of hSTAT2 KI mice. In addition, hSTAT2 KI model requires a special strain of ZIKV (DakAr41525), which has

been adapted to replicate in mice. This strain belongs to the African lineage of ZIKV, which is very distantly related to the epidemic Asian ZIKV lineage (including the Paraiba 2015 strain used in our study) and might not recapitulate some aspects of ZIKV pathogenesis in the MRS that are specific for the strains of epidemic Asian lineages. Another described immunocompetent mouse model of ZIKV pathogenesis (PMID: 29844387) also relies on the virus strain of African lineage (MR766). Considering that strain MR766 has a very long passage history in the brains of suckling mice (>150 passages), this most likely caused changes in the cellular tropism of the virus, compared to natural ZIKV isolates. The MR766 virus also requires unnatural (intravenous) route of infection with a very high dose of the virus to facilitate virus seeding of mouse organs, which also can change a natural course of ZIKV infection of the MRS. Thus, analysis of ZIKV interaction with MRS of immunocompetent mice would require numerous adjustments to study design and subsequent validation of a new model for sexual transmission of ZIKV. This does not seem feasible, considering that we have already reached the limit on manuscript length, and should be a focus of future studies.

B - It is possible to make immunocompetent mice susceptible to ZIKV by treating them with anti-IFN α /b antibodies just before virus infection (PMID: 27798603). However, studies demonstrated that in this 'semi-immunocompetent' model, ZIKV strains of Asian lineage do not replicate in the MRS very efficiently as compared to African strains (PMID: 27798603). Since insertion of heterologous sequences (scr and miRNA targets) into the genome of Asian ZIKV strain resulted in a considerable attenuation of ZIKV fitness even in AG129 mice, we anticipate that genetically modified viruses used in this study (Asian strain) will not be able to reliably infect MRS of immunocompetent mice treated with anti-IFN α /b. This also suggests that in order to achieve consistent results in immunocompetent mice, we will have to develop and re-test a new set of ZIKV clones using African ZIKV strains such as DakAr41525 or MR766. In addition, there are many variables inherently associated with anti-IFN α /b antibody treatment of mice (that would potentially complicate our comparative virus analyses and make it difficult to interpret them): (i) a route of antibody administration; (ii) an optimal dose of antibodies to saturate the target (suggested to be a very large bolus injection, ref. PMID:17115899); (iii) levels of antibody extravasation and penetration (if any) in a given tissue compartment ; (iv) timing of antibody administration and repeated dosing before and/or during ZIKVs infection (arbitrarily chosen in one report (see PMID: 28068342) at day -1, +1, and +4, which resulted in a high variability of observed outcomes); (v) target distribution, saturation, and half-life of injected antibodies in the MRS (especially in the immunoprivileged testis and epididymis (see PMID:2372399 and 24954222); (vi) efficiency of binding and interferon receptor neutralization by antibodies in different cellular and structural compartments comprising the testis and epididymis. Thus, given a complexity of our experimental design with a large number of different viral constructs to be compared in a consistent manner and monitoring kinetics of their replication in the MRS, we could not afford to control for all potential variables mentioned above. As a side note: ZIKV tropism for the cells of testis and epididymis observed in a "semi-immunocompetent" mouse model (PMID: 27798603) resemble those that we observed in AG129 mice in this and in our previous study (PMID: 30559387). These observations strongly suggest that (at least with respect to mouse models of ZIKV infection) intact interferon signaling is only effective in preventing a systemic ZIKV dissemination from the site of infection, but it might not be sufficient in preventing ZIKV replication in the organs of MRS after the virus is seeded there with viremia. This justifies the use of AG129 mice for studies of ZIKV infectious events that occur after systemic spread with viremia and seeding of the virus into the target organs.

C - Close evolutionary relationship to humans makes non-human primates (NHP) an ideal model organism for evaluation of ZIKV tropism in the MRS. However, such studies would require infection and sacrifice of a considerable number of animals, which would be difficult to justify for apparent ethical

reasons. For instance, several time course studies will be needed only for the control 2xscr virus to determine an appropriate infectious dose and meaningful time intervals for organ collection, which should be followed by another study that uses different recombinant ZIKV constructs. It just would be not feasible to obtain that many animals, especially during the current SARS-CoV-2 pandemic when most of available NHP are used for studies of SARS-CoV-2 pathogenesis. Moreover, the organs of MRS of NHP demonstrated a high degree of variability with regard to susceptibility and ability to establish persistent ZIKV infection (PMID: 32761051, 31597777). Addressing this issue would require an additional increase in the number of experimental animals in order to obtain statistically significant results. As a side note: we plan to study ZIKV infection of the MRS of NHPs by developing a novel barcoded ZIKV library, which could potentially reduce the number of required experimental animals. Such new virus library would only contain synonymous polymorphisms in the open-reading frame of ZIKV, which will be different from the ZIKV-lib/n11 virus library used in this study. We believe that the ZIKV-lib/n11 virus library is not well suited for the studies in NHPs since the modifications present in the 3'NCR of ZIKV-lib/n11 (see **Fig. 5**) may impose significant attenuating effects on ZIKV fitness in a primate host.

Other comments:

Line 122 – authors note that 2x141(T)mut was not detected in testis, but do not show any data points in Fig 1D for this virus or do they state data not shown. And also vice versa for 2x202(T)mut in epididymis (Fig 1E).

Response 1.2:

In the legend for **Fig. 2** we stated: “If all virus samples from the organ remained stable, no additional notations were made to the virus name.” This implies that in **Fig. 1D** and **Fig. 1E** all data points for 2x141(T) and 2x202(T) correspond to viruses that have stable genome (miRNA targets). We have also considered adding notation ‘stb’ to the virus name in cases where all viruses recovered from the organ have stable genomes [i.e. virus 2x141(T) for **Fig. 1D** and 2x202(T) for **Fig. 1E**]. But we concluded that this approach might create more confusion for the readers. For instance, if all virus samples recovered from the organs were proven to be stable, the readers could be confused why there are no data points on the graph with notations ‘mut’ (i.e. 2x141(T)mut for **Fig. 1D**). Also, for the reasons of consistency, we would have to add notation ‘stb’ for all 2xscr viruses in **Fig. 1** and **3**, which would also be confusing and unnecessary.

Male mice were infected IP with high titers of virus, this is not a natural route of infection.

Response 1.3:

We agree that normally ZIKV infects its host via a bite of an infected mosquito, and that intraperitoneal (IP) infection with a dose of 10^6 pfu cannot be considered a natural route of ZIKV exposure. However, during replication in mosquitos, engineered viruses can accumulate mutations in the miRNA targets, which may dramatically reduce the heterogeneity of barcoded viruses. Most importantly, it is impossible to control the amount of virus that each mosquito delivers to the mouse, making this experimental approach impractical for studies of virus pathogenesis. Therefore, needle inoculations were used for mouse experiments, which is considered to be a standard practice for infection of experimental animals with arthropod-borne viruses.

ZIKV pathogenesis studies typically use natural virus isolates and utilize subcutaneous route of virus infections using dose ranges between 10^3 to 10^5 pfu. However, in this study we rely on the genetically modified clones of ZIKV, which are attenuated compared to either natural ZIKV isolates (PMID: 27555311, 27198478) and/or clone-derived ZIKV that do not carry the heterologous sequences in the 3'NCR (PMID: 30559387). In order to compensate for these attenuating effects, we use a high infectious dose (10^6 pfu/mouse) and utilize a more permissive intraperitoneal (IP) infectious route. Previously, we

demonstrated that replication pattern of miRNA targeted viruses in the testis and epididymis of AG129 mice following IP infection with 10^6 pfu of the virus was very similar to those observed for natural ZIKV isolates in immunocompromised and immunocompetent mice infected subcutaneously (see PMID: 30559387, and 27798603, 28199846, 28680856, 27884405, 29378173). Finally, a high infectious dose and reliance on IP (instead of subcutaneous) inoculation avoids the genetic bottlenecks associated with ZIKV replication in the subcutaneous space, which can negatively impact the heterogeneity of barcoded ZIKV in the mouse serum and subsequent seeding of these viruses in the organs of MRS. All these considerations were important prerequisites for our study.

Change in the manuscript:

To clarify our position, the following sentence was added to the text (Line 659): “High virus dose and IP route of inoculation were chosen to compensate for a moderate attenuating effect associated with insertion of heterologous sequences (such as miRNA targets) into 3’NCR of ZIKV genome^{18,53}.”

Reviewer #2 (Remarks to the Author):

However, the use of immunocompromised mice is a significant limitation of the study. As a main concern, an overall consideration of the limits of this model to decipher the sexual transmission of ZIKV in human and non-human primate models is missing. Moreover, some figures’ interpretations should be clarified.

Major comments:

- The strong deleterious effect of ZIKV on the male reproductive tract of immunocompromised mice such as AG129 drastically differs from that in ZIKV-infected men and NHP (both ex vivo and in vivo), in whom organ morphology is generally preserved despite the infection and semen parameters only transiently and mildly impacted. This needs to be clearly stated in the manuscript. Thus, in immunocompetent hosts such as human or NHP, a number of cell types within the MRT may not be readily permissive to ZIKV infection. This might lead to important differences in tropism and hence in the contributions of male genital cells/organs to ZIKV shedding into semen. The mammalian testis is an immunoprivileged organ with immunosuppressive properties as well as weakened IFN responses (see Le Tortorec et al 2020 Physiological reviews). In those preserved immunological conditions, the testis might represent a more favourable environment for ZIKV replication and persistence, and hence play a more prevalent role than in this immunocompromised animal model. Conversely, a functional IFN system may considerably reduce the infectivity of the epididymis and in turn decrease its relative contribution to ZIKV seminal shedding in semen. Therefore, the use of an immunocompromised model such as AG129 mice can severely bias the relative contribution of male genital organs to ZIKV shedding into the semen, and this needs to be properly acknowledged as an important limitation for extending those conclusions to humans. The authors should therefore discuss the limits of their model regarding the prediction of trafficking and persistence of ZIKV in human and NHP male genital tract in the light of these elements. For the same reasons, the title of the article and abstract should specify that this work was performed in immunocompromised mice that were acutely infected.

Response 2.1:

We agree with reviewer #2’s comments regarding limitations of AG129 mouse model of ZIKV. Please see our **Response 1.1** to reviewer #1’s comment.

- Figure 1E and 1F, lines 163-169 : the authors state that the absence of apparition of escape mutation of 2x202 in the epididymis and vas deferens over time indicate that the excurrent transport of ZIKV from testis is quite inefficient. Bearing in mind the timing for spermatozoa excretion from testis into semen

and that escape variants are detected in the testis only from 12 dpi, they cannot exclude at this stage the requirement for longer kinetics to seed infection in these organs.

Response 2.2:

Although here we showed that efficient ZIKV migration from testis to epididymis occurs as early as 10 dpi (see **Fig. 5E**), we agree with reviewer #2 that longer kinetics of ZIKV infection are needed to determine whether ZIKV virions produced in the testis can reach ejaculate and infect female partner. Unfortunately, as shown previously, male mice start to exhibit neurological symptoms beyond 17 dpi (which are caused by ZIKV replication in the CNS; See **Fig.6** and PMID: 30559387) and have to be euthanized. We believe that we have addressed this limitation and discussed alternative strategies that can mitigate the survival issue of infected male mice: see discussion lines 514-556, the paragraphs starting with “Previously, using miRNA-targeted viruses...”

- Another important point is that the seminal shedding examined in this work only corresponds to acute infection, and not persistence over several weeks or months as reported in men up to over one year. As expected and mentioned by the authors, mating occurred in the first days of contact between female and 7 day infected males, so at best ejaculates reflect d12-infected mice. While this issue is acknowledged in the discussion, it needs to be clearly indicating from the title onwards.

Response 2.3:

Upon reviewer #2’s suggestion we modified both the title and abstract to emphasize that presented study analyzes the role of different parts of MRS during early phase of ZIKV infection.

Knowing the physiology of the MRT, it is indeed unlikely that the testis would represent a major contributor to virus release in semen during the early stage of the infection, and the results of this work are therefore not surprising in that respect. The main contribution of the testis to semen is the production of sperm, which take about 35 days in the mouse (74 days in men) to form from spermatogonia, which may represent the primary initial targets of ZIKV in the seminiferous tubules. Testicular spermatozoa then transit through the epididymis for about 10 days in mammals, before being released in semen. Seminal fluid is at >90% constituted from the secretions of the prostate, seminal vesicles and epididymis. Thus one expect testis contribution to semen infection to be mostly through infected cell release, rather than through secretions. This could play a role in the long term rather than during the acute stage. Moreover, the results suggest that the timing of virus spillover across accessory glands could differ according to their respective anatomical distances.

Response 2.4:

We disagree with reviewer #2’s assumption that the time required for production/maturation of sperm cells in the testis is similar to the time needed for ZIKV to generate infectious particles capable of migration from testis into the epididymis with efferent flow of sperm. In cell culture, the cells infected with ZIKV start secretion of infectious particles within 24 hours post inoculation (see new **Fig S4** as an example of ZIKV replication in cell culture), suggesting that ‘transportable’ ZIKV can be generated in the testis quite rapidly (within 10 days post mouse infection - see **Response 2.2**). In addition, it would not be correct to simply assume that the time of ZIKV transition through epididymis is comparable (or greater) to the transition time of spermatozoa. Since virion is much smaller than spermatozoa it is possible that virions could traverse epididymis faster than spermatozoa. It was this study that demonstrated that these times are indeed comparable.

We also cannot accept reviewer #2’s conclusion that “testis contribution to semen infection to be mostly through infected cell release, rather than through secretions”. For instance, it was demonstrated that ZIKV virions in semen exist mostly in cell-free form (PMID: 30070988). Moreover, the term ‘release of infected cell’ is quite unclear. If by this reviewer #2 means ZIKV genomes/maturing virions located inside the spermatozoa (which were produced in the testis then become part of the semen), then such

material would not be infectious, and therefore cannot contribute to the infection of the mating female partner (it is therefore irrelevant to sexual transmission of ZIKV). However, if by term 'release of infected cell' reviewer #2 means infectious ZIKV particles that are "released by infected cell" (of testis), then there would not be much of a difference between "infected cell release" of the testicular cells and the "secretions" of accessory glands, since in both cases the infectious virus is produced by the cells that just happen to be located in different organs/parts of MRS.

Considering these factors, we believe that it would not be scientifically correct to infer relative contribution of different parts of the MRS in shedding of ZIKV into the semen using only the knowledge of the MRS physiology, and that our study provides important mechanistic insights needed for a better understanding of ZIKV sexual transmission.

- Line 81, information about the physiological role of mir-202 in the seminiferous tubules should be added, similarly to mir-141, and expression in testis interstitial cell discussed. Moreover, the authors should comment on how the miR-engineered virus may affect organ functions such as epithelial cell barrier integrity in the epididymis and spermatogenesis in the testis (e.g by "sponging" the miRNA from their physiological targets)?

Response 2.5:

A -The precise physiological role of mir-202-5p in the testis (not to be confused with mir-202-3p) is unknown. However, this information is not very relevant to our study, since expression of mir-202-5p is confined only to cells located in the seminiferous tubules of the testis. We mentioned the role of mir-141-3p since it allowed to connect the pattern of mir-141 expression with a specific cell morphology. Cells of this type are abundantly present in many organs of MRS and can be targeted by ZIKV.

B -We have previously demonstrated that in the testicular interstitium ZIKV primarily targets CD206/mir-511-3p-expressing macrophages (PMID: 32614902), which do not express mir-202-5p (see **Table S2** in PMID: 20605486). These findings are in agreement with our previous observation that insertion of targets for mir-202-5p into ZIKV genome does not inhibit ZIKV infection [virus 2x202(T)] of testicular interstitial cells, which occurs around 3 dpi (See **Fig. 2, 3** in the PMID: 30559387). ZIKV virions produced by cells of testicular interstitium cannot reach epididymis because they do not have access to the lumen of seminiferous tubules (for that, ZIKV needs to bypass the blood-Sertoli cells barrier and replicate inside the cells of seminiferous tubules). In contrast, infection of cells located inside the seminiferous tubules occurs considerably later (begins after 6 dpi; PMID: 30559387, 32302362). Therefore, targets for mir-202-5p can only inhibit this step of ZIKV infection of the testis.

C - Sequestration of cellular microRNAs in the infected cells due to a 'sponge effect' can theoretically alter physiological functions of an infected organ. However, ZIKV infection and replication in the target cells inflicts very dramatic changes onto cell physiology (gene expression, membrane architecture and functions, et cetera), eventually leading to cell death due to viral or immune system-mediated factors. In this context, changes in cell physiology inflicted by sequestration of only one type of cellular microRNAs would have a rather negligible effect compared to overall pathogenic changes associated with viral infection.

Change in the text: following sentence was added to the text Line 86: "The precise role of mir-202 in testicular physiology has yet to be described."

- How do the authors explain that miR-202 stb virus abrogate replication in the testis, although mir202 does not target macrophages, an important infected cell type in the testis?

Response 2.6.

Please see our point B in **response 2.5**

To clarify our position we modified the sentence “Previously, we showed that ZIKV genome targeting for mir-202 effectively restricts ZIKV replication in the testes,” to (Line 94) “Previously, using 4-6 weeks-old AG129 male mice that lack type I and type II interferon receptor (IFNR) genes, we showed that ZIKV genome targeting for mir-202 effectively restricts ZIKV replication in the cells located inside of seminiferous tubules, which blocks ZIKV migration from testis to epididymis²¹.”

- Fig3 E-H: this figure should show the level of mir 141 in the testis and other organs to make sure that lack of transmission did not reflect absence of MRT infection by the variant in this experiment. Why is this data missing? The indirect measurement of ZIKV mir-141 in semen through virus load measure in female mice only confirms the lack of transmission of this virus in acute stage but does not demonstrate MRT infection (eg testis) in those mice.

Response 2.7

In the legend for Figure Fig. 3E-3H we specified that ‘Reproductive organs were analyzed only for those males whose female mating partners developed detectable ZIKV viremia’, which made possible analysis of miRNA targets stability in the recovered viruses. This was done to infer a conclusion regarding a potential site of origin of transmitted ZIKV genomes within the MRS of male mice, which were then transmitted to the female. Viruses isolated from females can be sequenced to determine stability of miRNA target (and scr) sequences, and then compared to viruses from MRS of male mice. Since 2×141(T) virus has not been isolated from any of mating females, we did not have a reason to do the analysis of the 2×141(T) virus in the MRS of male mice at 17 dpi (the time when male mice were sacrificed), since correlations between transmitted 2×141(T) genomes and MRS sites would not be possible for these mice. Stability and corresponding titers of 2×141(T) virus in all MRS organs at 17 dpi were presented in Fig. 1D-1G and Table S1. They show high viral load in testis of all analyzed mice (n=5). We, however, noticed that in the original version of Fig. 1D-1G not all data points for 17 dpi were depicted on the graphs (although the mean values were depicted accurately). It happened due to a peculiarity of the Prism 8 program, which hides some data points for the last time interval (17 dpi) in the staggered plot configuration. We corrected this error in the revised version of the Fig.1, and now all data points are accurately depicted. Please note that all 5 mice inoculated with 2×141(T) virus have high titer of the virus in the testis at 17 dpi.

- Figure 5E: even if no common barcodes were detected in between testis and vas deferens, it could be informative to analyse the presence of common barcodes in between the testis and the SV/P.

Response 2.8

Upon reviewer #2’s request we analyzed distribution of common barcodes between the testis and the SV/P and compared them to the distribution of common barcodes in the testis-brain and testis-epididymis organ pairs. As expected, we did not detect significant differences ($p>0.05$, one-way ANOVA) among mean combined frequencies of common barcode (CFCB) values for testis-brain and testis-SV/P pairs, while mean CFCB for testis-SV/P values were significantly lower than CFCB for testis-epididymis ($p>0.0001$; one-way ANOVA). These new findings were presented in the updated version of Fig. 5E and were discussed in the modified Results paragraph (Lines 566-583).

- Fig 6 and lines 386-394: it is striking that the majority of the strains detected in the ejaculate arise from the vas deferens, suggesting that viruses are flushed out from epididymis tail/vas deferens during ejaculation, where sperm are stored. Can the authors assess whether the viruses detected in the ejaculate are bound to sperm?

Response 2.9

We agree with reviewer #2's interpretation of the results that virus in ejaculate is most likely flushed out from epididymis tail. We did not assess whether ZIKV in the ejaculate is bound to the sperm cells or exists as free-floating particles. However, previous studies have demonstrated that ZIKV in the semen exists mostly in cell-free form (PMID: 30070988). Our results are consistent with a model where free-floating ZIKV particles are generated by the cells of epididymal epithelium and get secreted into the lumen of epididymal tubules. The virus subsequently passes through vas deferens and becomes part of the ejaculate.

Changes in the text:

To clarify our position the following sentences were added to the Discussion Lines 909-914: "Together, our results demonstrate that ZIKV that was sexually transmitted between 7-11 dpi was primarily produced by the cells of epididymal epithelium. These types of cells may secrete Zika virions into the lumen of epididymal tubules in a cell-free form²⁰, and these virions, in turn, are flushed out of the epididymis during the emission phase of the ejaculation (see Fig. 5C for schematics) and eventually expelled into the ejaculate and become sexually transmitted"

Specifics comments:

- Line 41: the choice of references is questionable as there are more comprehensive/recent reviews and articles to quote.

Response 2.10

References in the Line 41 (Line105 in revised manuscript) have been updated.

- Line 57-60: In Osuna et al, persistent (28 dpi) infected cell foci were reported in testis, prostate and seminal vesicles. Therefore it is not correct to state that only testis and epididymis were reported in NHP as being persistently infected. Persistence of ZIKV RNA was also detected in vasectomized men, ruling out epididymis and testis as a source in these cases. Furthermore, the MRT infection across NHP studies is overall very low, with discrepancies as to the target organs and little seminal shedding. Thus, caution must be exerted and no definite statements should be made from this model.

Response 2.11

We acknowledged the challenges with interpretation of the result of non-human primates' studies.

Change in the manuscript:

We substituted original sentences "Experimental infection of various species of new- and old-world primates demonstrated that ZIKV replicates in the testes, epididymis, prostate, and seminal vesicles, but the virus can establish persistent infection only in the testes and epididymis^{21,23-25}. Collectively these observations suggest that testes and/or epididymis might be the primary source of ZIKV in the semen" with the following text lines 60-66 "Studies in NHPs tend to indicate a higher load and longer persistency of ZIKV RNA in the testis and epididymis, as compared to other MRS organs^{29,27}. This incriminates testis and/or epididymis as a possible source of ZIKV in the semen, which is consistent with reports of a prolonged detection of ZIKV-positive spermatozoa in humans³⁰. However, a high variability of the results obtained in NHPs makes it challenging to identify the precise role that each individual MRS component plays in ZIKV infection. Moreover, ..."

- Lines 60-61: the authors may reinforce the message of a persistent infection in the testis and/or epididymis with references to studies that demonstrate the long term shedding of ZIKV positive spermatozoa in human semen.

Response 2.12

Following sentence was added to the text lines 63-64:” ... which is consistent with reports of a prolonged detection of ZIKV-positive spermatozoa in humans³⁰.”

- Figure 1E, D: The authors should comment on the fact that levels of mir-202 stb virus in the testis at day 6 are similar to that of other viruses (and decrease thereafter), whereas mir-141 stb virus in the epididymis at day6 is several log lower than others, but increases over time. The explanation that the increase of mir-141 in the epididymis is due to transport of virus from the testis could appear challenged to the reader by the fact that this is not observed for ZIKV-NS3m and scr, which also replicate in the testis and show even more increase replication in testis than mir-141 over time. Differences in viral load threshold likely account for this but this needs to be explained. To consolidate the idea that excurrent transport plays a role, the authors could refer to their previous published data showing that at 12 dpi ZIKV positive antigen immunoreactivity for mir-141 stb was exclusively found in luminal cells. Overall, this part of the ms is a bit confusing and hard to follow.

Response 2.13

Previously we and others showed that ZIKV infection of testis and epididymis begins with infection of the cells located in the interstitium of both organs (PMID: 32614902, 30559387, 28680856). After that, infection progresses to the seminiferous tubules (testis) and epididymal tubules (epididymis). Available data indicate that ZIKV infection of the epididymal epithelium occurs considerably earlier than infection of seminiferous tubules. For instance, between 5-8 dpi ZIKV was detected primarily in the interstitium of the testis (PMID: 32302362, 28261663, 30070988), and not in the seminiferous tubules. However, at this time interval ZIKV was already found infecting epithelial cells of the epididymis (PMID: 28261663, 30070988, 30337585).

Since 2×202(T) (202-stb) can infect cells of testicular interstitium (but not seminiferous tubules [see **Response 2.5**]), replication of this virus in this testicular compartment should not be significantly different compared to 2×scr or other viruses. This can explain why the titer of the 2×202(T) in the testis was similar to 2×scr at 6 dpi. Subsequently, viruses which do not have targets for mir-202 start to invade seminiferous tubules (includes virus 2×202(T)mut), thus explaining the increase in virus titers between 6 and 9 dpi.

In contrast, by 6 dpi the 2×scr virus can already infect epithelial cells of epididymis. However, 2×141(T) virus [see 2×141(T)-stb] cannot progress from epididymal interstitium to epithelial tubules. This is why we observed a low titer of 2×141(T)stb in this organ. By 9 dpi, ZIKV (2×141(T)stb) infects seminiferous tubules of the testis, and only at this time this virus can be secreted from the testis to the epididymis. This can explain why there was a gradual increase in 2×141(T)stb titers observed between 6 and 17 dpi.

Change in the text:

To clarify this position and to simplify interpretation of experimental results depicted in **Fig 1**, the following sentences were added to Results (Line 120-125): “Seeding of ZIKV in the testis and epididymis begins with infection of the cells located in the interstitium of both organs^{16,21,23}. Subsequently, infection progresses to the cells located inside the seminiferous tubules (expressing the mir-202) or epididymal epithelium (expressing the mir-141). It was shown that in the epididymis, progression of ZIKV infection from the interstitium to epithelium occurs ~2-4 days earlier than progression of infection to seminiferous tubules in the testis^{20,43-45}”.

- The authors suggest that epithelial cells are the main infected cell types in the epididymis and seminal vesicles/prostate. What then explains the much lower infection levels of seminal vesicles/prostate versus epididymis, especially if those organs are additionally seeded with virus coming from the epididymis?

Response 2.14

We can only speculate what might be causing differences in the ZIKV infection rate (and in titers) between epididymis and SV/P. Although epididymis is not considered a classical immune privileged organ (it does not promote survival of allografts), it nevertheless demonstrates some specific immunoregulatory mechanism that ensures immune tolerance toward autoantigens expressed on the surface of maturing spermatozoa (reviewed in PMID: 33117332). Epithelial cells of seminal vesicles and prostate do not come in contact with these autoantigens, and therefore, inflammatory responses triggered by ZIKV infection may be more robust (as compared to epididymis), resulting in better control of ZIKV proliferation in these organs (decreased ZIKV infection and ZIKV titers in the SV/P).

In this study we did not specifically analyze whether ZIKV generated in epididymis can infect epithelial cells of the SV/P (seeding of the SV/P by the epididymis-derived virus). Nonetheless, our data suggest that even if this infection could occur, this seeding mechanism would probably be less efficient compared to hematogenous seeding of ZIKV in the SV/P, since majority of the ZIKV barcodes detected in the SV/P were not detected in the epididymis (**Fig. 5F**). Furthermore, we believe that the detection of ZIKV genomes that have specific molecular signatures of the epididymal origin (either barcodes or deletions) in the SV/P likely comes from the contamination of SV/P with parts of the ampulla during dissection of these organs (See **Fig 5C, S3**). Ampulla stores sperm cells and, therefore, it might also contain ZIKV particles generated in the epididymis.

- Figure 3C, the measure of infectious titer in female serum would be more convincing to demonstrate the equivalent virus infectivity properties of the viral clones.

Response 2.15

In the experiment depicted in **Fig. 2B** the assessment of sexual transmission efficiency was conducted using detection of ZIKV antibodies in serum of the female. Therefore, we concluded that it would be only appropriate to analyze whether viral clones induce seroconversion in similar proportion in female mice after intravaginal inoculation (**Fig.2C**). We agree that expression of mir-141 by the cells of FRS might have contributed to reduced infectivity of 2×141(T) virus to female mice. We performed a detailed comparison of 2×scr and 2×141(T) in the serum and FRS of AG129 female mice after intravaginal inoculation. Results of these studies are provided in a new **Fig. S7** and in the description of this study, which is given after the legend to **Fig. S7** in supplementary materials. These new results demonstrate that, at 6 dpi, 2×141(T) virus attains significantly lower titers (as compared to 2×scr) in the serum and in some parts of the FRS. The observed discrepancies in the effects of mir-141 targeting on ZIKV infectivity to different parts of FRS and on seroconversion rate of female mice (**Fig. 3C**) precluded us from making a definitive conclusion regarding the role of mir-141 targets in infectivity of 2×141(T) virus to female mice via sexual transmission. These results prompted us to perform the semen shedding studies, which allows us to omit the female infection step altogether (**Fig. 4**).

- Line 238, there is a mistake in the figure reference, 3E instead of 3B.

Response 2.16

Mistake is corrected.

- Line 303, edition mistake after the references.

Response 2.17

Mistake is corrected.

- Lines 325-327: a reference that could be added, in support of the authors' hypothesis on the existence of tissue-specific viral variants in the MRT, is one regarding SIV organ-specific sequences/phylogeny in the MRT of macaques (Houzet et al, J Virol 2018), which is also showing virus intermingling between

testis and epididymis. This could be commented on in the discussion in order to extend the authors' observations on ZIKV dissemination in the MRT to other viruses and models.

Response 2.18

The reference to Houzet et al was provided in the modified sentence (lines 338-341): "We reasoned that similar to a simian immunodeficiency virus (SIV) infection of MRS in cynomolgus macaques⁴⁸, the hematogenous dissemination of ZIKV from the infectious site should be followed by virus seeding and replication in the organs of the MRS^{21,23}."

The sentence (lines 524-526) ". For example, SIV has been shown to originate from multiple genital organs of NHPs and migrate through the testis-epididymis-vas deferens axis during a long-term chronic infection" was also added to discussion to indicate that virus migration between testis and vas deferens can probably occur for ZIKV, but likely requires longer than 11 dpi.

- Lines 364-365: it would have been interesting to test further time points using 17 day-infected male mice to determine whether testis-borne viral strains travel out of the epididymis, with or without ejaculation.

Response 2.19

Please see our **Response 2.1**.

Analysis of ZIKV replication in the MRS of male mice post 17 dpi is very important and will be addressed in future studies. Unfortunately, male mice start to exhibit neurological symptoms beyond 17 dpi (which are caused by ZIKV replication in the CNS; See **Fig.6** and PMID: 30559387) and had to be euthanized. We believe that we have addressed this limitation and discussed alternative strategies that can mitigate the survival issue of infected male mice: see discussion, two modified paragraphs (lines 527-556)

- Lines 449 and 471: please precise "during the acute stage" for epididymis as a main contributor of sexually transmitted ZIKV in immunocompromised mice.

Response 2.20

We modified both sentences to precisely specify the timeframe of sexual transmission events that were analyzed in this study.

Sentence1 (line 463-465): "The results of two independent yet complementary approaches used in this study identified the epididymis as a leading contributor of the sexually transmitted ZIKV genomes during the early phase of infection (7-11 dpi)".

Sentence2 (486-487): "Together, our results demonstrate that the ZIKV that was sexually transmitted between 7-11 dpi was primarily produced by the cells of epididymal epithelium."

- Line 524, there is a misspelling for the word "stability"

Response 2.21

Misspelling is corrected.

- Line 528, there is a misspelling for the word "transmitted"

Response 2.22

Misspelling is corrected.

- Line 538: the immunocompetent mouse model developed by Robinson et al (Nature communication, 2018) could also be of interest for further MRT studies.

- Line 485: an important point to discuss here is how ZIKV target cells in the testis compare between immunocompromised and immunocompetent mice models (Robinson et al, Nature Communications 2018) and human testis (Matusali et al, JCI 2018).

Response 2.23

We agree that the model Robinson et al have described is an interesting immunocompetent mouse model of ZIKV infection. We added a reference to this model (PMID: 29844387) to our discussion of alternative approaches that can be used to validate findings of our paper (see line 555). We also added reference to PMID: 29844387 to the discussion of general aspects of ZIKV infection of the testis of mice. We modified our original sentence "...replicate efficiently in testes of mice that have permanent (genetic) or transient (anti-IFNAR-I antibody treatment-induced) deficiency in IFN signaling pathways" to (line 502-503) "Between 7 and 17 dpi, natural ZIKV isolates replicate efficiently in the testis of both the immunodeficient and immunocompetent mice^{16-19,21,25,52}". Unfortunately, in their paper Robinson et al did not analyze ZIKV infection of epididymis, and it remains unknown whether mouse epididymis (and epididymal epithelium) can be infected with MR766 ZIKV strain following intravenous inoculation. Considering that (i) in our paper we did not obtain any evidence incriminating testis as a source of sexually transmitted ZIKV genomes, and (ii) due to manuscript size limits, we do not find sufficient justification for further discussions of findings of Robinson et al as well as Matusali et al (PMID: 30063220) in our manuscript.

- Line 560: mention here sexual transmission by a vasectomized man outside acute stage (Arsuaga et al, lancet Infect Dis 2016)

Response 2.24

We added the reference to Arsuaga et al and modified sentence in Line 560 by adding the following phrase (See line575): "which can potentially occur outside the acute phase of ZIKV infection³¹."

- Line 635, there is a misspelling for the word "strain"

Response 2.25

Misspelling is corrected.

- Line 709, there is a misspelling for the word "contrast"

Response 2.26

Misspelling is corrected.

- Line 753, "primary" should be "secondary".

Response 2.27

Mistake is corrected.

Reviewer #3 (Remarks to the Author):

Major concerns:

To further support the role of the epididymal epithelium as a source of the ZIKV genomes/virions secreted into the semen:

i) Is there any possibility to only dissect the target infected tissue and sort these cells by flow cytometry using a virus antibody or targeting a GFP-like viral construct?

Response 3.1:

The experimental approach based on analysis of ZIKV genomes derived from flow cytometry-selected cells has a number of critical limitations for genome tracing studies:

1) Several ZIKV clones capable of expressing GFP have been developed (PMID: 27471954, 29022864, 31947825). However, all these clones demonstrated significant instability of GFP gene and showed drastic reduction of viral replication fitness. These factors limit application of such clones to cell-culture work and make them impractical for prolonged *in-vivo* studies (that take more than 2-3 days post infection).

2) It might be possible to select ZIKV infected cells from organs of MRS using ZIKV-specific antibodies. However, cell-sorting cannot guarantee that 100% of infected cells will be captured and used in the subsequent sequencing analysis. This implies that there would be considerable probability of losing some organ-specific genomes. Moreover, at the time of organ dissection (which occurs at 10-11 dpi) a considerable number of ZIKV-infected cells might be already dead due to ZIKV cytotoxicity or immune system-mediated clearing of infected cells. This implies that organ-specific genomes produced by these (dead) cells also will be omitted from the analysis aimed at correlating ZIKV genome signature in the given organs with ZIKV genome signature in the ejaculate. Together, these factors preclude us from reaching a definitive conclusion regarding the role of organ-specific cell types in production of sexually transmitted ZIKV genomes using a cell-sorting approach.

ii) Use this tissue to extract viral RNA and perform negative-strand-specific RT-PCR to confirm viral replication. I would do the same on the other tissues/organs of the MRS

Response 3.2:

In **Fig. 2** we presented/analyzed distribution of ZIKV infected cells in the organs of MRS. For this analysis we used an anti-Zika virus antibody specific for NS2B protein of the ZIKV. The NS2B is a non-structural viral protein of the virus, which can only be generated/detected in the host cell during active virus replication, i. e. presence of NS2B immunoreactivity directly confirms active viral replication in the host cells. Therefore, we believe that the experiment suggested by Reviewer #3 (detection of negative-strand of virus RNA by RT-PCR to confirm viral replication) is redundant.

Change in the manuscript:

To clarify our position the following sentence was added to the manuscript (Line 775, Materials and Methods): **The NS2B is a non-structural protein that can be detected in the host cell during active virus replication.**

iii) More experiments to support this key claim could be address on DC2 cells (immortalized mouse distal caput epididymal epithelial cell line).

Response 3.3:

Upon Reviewer #3's request we purchased DC2 cells and compared replication kinetics of a control (2xscr) and miRNA-targeted viruses [2x202(T) and 2x141(T)] in these cells. Compared to 2xscr virus, replication of only 2x141(T) virus (but not the 2x202(T) was significantly attenuated in the epididymal epithelial DC2 cells (see new **Fig. S4**). These results demonstrate that epididymal epithelial cells can support ZIKV replication, which is selectively attenuated by mir-141 genome targeting.

Change in the manuscript:

The following sentences were added to the text (Line 168-171): **"In addition, growth of 2x141(T) virus in immortalized mouse distal caput epididymal epithelial cell line DC2 was significantly attenuated compared to 2xscr virus (Fig, S4), confirming that mir-141 targeting strongly attenuates ZIKV infectivity for epididymal epithelium²¹."**

Minor comments:

What about the mir-141 in the female mice?

Response 3.4

Expression of mir-141 by the epithelial cells of FRS might have contributed to the reduced infectivity of 2×141(T) virus to female mice (**Fig. 2B**). We performed a detailed comparison of 2×scr and 2×141(T) in the serum and FRS of AG129 female mice after intravaginal inoculation. Results of these studies are provided in a new **Fig. S7** and in the description of this study, which is given after the legend to **Fig. S7** in supplementary materials. These results demonstrate that at 6 dpi the 2×141(T) attains lower titers in the serum and in some (not all) parts of the FRS (as compared to 2×scr). The observed discrepancies in the effects of mir-141 targeting on ZIKV infectivity to different parts of FRS and on seroconversion rate of female mice (**Fig. 3C**) precluded us from making a definitive conclusion regarding the role of mir-141 targets on infectivity of 2×141(T) virus to female mice. These results prompted us to perform the semen shedding study (**Fig. 4**), which allows us to omit the female infection step altogether and provides a simpler model for analysis of the roles of mir-141-expressing cells in the seminal shedding of ZIKV.

Reviewers' Comments:

Reviewer #1:

Remarks to the Author:

The authors have tried their best to address concerns raised by the reviewers. I am satisfied by the response although feel that the title should be modified and include 'immunocompromised mouse model' (something similar to shown below) to reflect the study.

Title: Epididymal epithelium propels early sexual transmission of Zika virus in the absence of interferon signaling in an immunocompromised mouse model

Reviewer #2:

Remarks to the Author:

The authors have addressed most of my concerns. The data obtained in this mouse model are strong and the authors are to be congratulated for this very nice piece of work. There are only a few remaining points regarding the discussion of their results with respect to other models and in humans that need to be addressed as detailed below.

- Line 35: need references of more recent/comprehensive reviews on sexual transmission of arboviruses (i.e. PMID: 32031468 and ref 11).
- Line 64 and discussion : Recent published data showing acute and prolonged release of productively infected testicular germ cells and epithelial cells in human semen (Matusali et al, JCI 2018; Mahé et al, Lancet Infect Dis 2020) need to be mentioned since they are pertaining to the origin of ZIKV in semen in humans and indicate testis involvement in addition to other organs, at least regarding cell-associated virus. Cell-associated virus is an important issue (not addressed in this study) when considering sexual transmission since cell-to-cell infection is usually much more efficient than infection with free viral particles. Thus even if in lower proportion than cell free virus, the role of cell-associated virus cannot be ignored (cell-associated ZIKV RNA was also detected in the semen of mouse models). Importantly, the ratio of cell-free versus cell-associated virus in semen may vary overtime.
- Authors' reply: "Sequestration of cellular microRNAs in the infected cells due to a 'sponge effect' can theoretically alter physiological functions of an infected organ. However, ZIKV infection and replication in the target cells inflicts very dramatic changes onto cell physiology (gene expression, membrane architecture and functions, et cetera), eventually leading to cell death due to viral or immune system-mediated factors. In this context, changes in cell physiology inflicted by sequestration of only one type of cellular microRNAs would have a rather negligible effect compared to overall pathogenic changes associated with viral infection": I disagree with the authors that the changes in cell physiology in their model, whether induced by the miRNA or viral replication, do not deserve attention. This certainly warrants caution as a potential limitation of the approach since any modification of the epididymis epithelial barrier induced by the virus in this immunocompromised model or by the E-cadherin targeting miRNA (E-cadherin playing an important role in the maintenance of the barrier integrity) may directly impact the shedding of viral particles especially in case of polarized shedding (eg allowing the release in the seminal lumen of viruses normally that might be shed only on the basal side of epithelial cells) and hence the results of this study. Although not obvious from the d9 pictures of the organs, if such a dramatic effect of ZIKV infection is observed in this mouse model that any impact of the miRNAs becomes irrelevant even at early time points, this needs to be acknowledged as a limitation to extrapolating the results to other models and to humans in whom such damage is not reported. Importantly also, the potential impact of the miRNAs on the cell physiology deserves to be mentioned in the context of the authors' vaccines containing miRNA targets.
- Line 524: miss the word "seminal" before "SIV"
- Comment on authors' reply: "ZIKV virions produced by cells of testicular interstitium cannot reach epididymis because they do not have access to the lumen of seminiferous tubules (for that, ZIKV needs to bypass the blood-Sertoli cells barrier and replicate inside the cells of seminiferous tubules)":

of note, the epithelium of the rete testis is considered to be less restricting than that of the Sertoli cells and may enable the leakage of testis interstitial virus and leukocytes into the seminiferous tubules and epididymis.

Response to the reviewers

We thank the reviewers for their insightful and helpful comments and recommendations.

Answers to reviewers' comments.

Reviewer #1 (Remarks to the Author):

The authors have tried their best to address concerns raised by the reviewers. I am satisfied by the response although feel that the title should be modified and include 'immunocompromised mouse model' (something similar to shown below) to reflect the study.

Title: Epididymal epithelium propels early sexual transmission of Zika virus in the 2 absence of interferon signaling in an immunocompromised mouse model

Response 1.1:

We appreciate reviewer's suggestion. However, the phrase from the original title 'absence of interferon signaling' already implies that an 'immunocompromised' system is used in the manuscript. In addition, we abundantly emphasized in the abstract and throughout the manuscript that an immunodeficient system was used for analysis of sexual transmission of ZIKV. Finally, Nature communication journal has 15-word limit on the length of the manuscript title. This effectively impedes our ability to additionally modify the title of manuscript without omitting other critical information which is currently present in the title.

Reviewer #2 (Remarks to the Author):

Major concerns:

- Line 35: need references of more recent/comprehensive reviews on sexual transmission of arboviruses (i.e. PMID: 32031468 and ref 11).

Response 2.1:

Requested references were added to Line 35.

- Line 64 and discussion: Recent published data showing acute and prolonged release of productively infected testicular germ cells and epithelial cells in human semen (Matusali et al, JCI 2018; Mahé et al, Lancet Infect Dis 2020) need to be mentioned since they are pertaining to the origin of ZIKV in semen in humans and indicate testis involvement in addition to other organs, at least regarding cell-associated virus.

Response 2.2:

References to Matusali et al, JCI 2018; Mahé et al, Lancet Infect Dis 2020 were added to sentence (Line 64). The sentence "This incriminates testis and/or epididymis as a possible source of ZIKV in the semen, which is consistent with reports of a prolonged detection of ZIKV-positive spermatozoa cells in humans³¹ was modified to acknowledge novel findings of Mahé et al.

Cell-associated virus is an important issue (not addressed in this study) when considering sexual transmission since cell-to-cell infection is usually much more efficient than infection with free viral particles. Thus even if in lower proportion than cell free virus, the role of cell-associated virus cannot be ignored (cell-associated ZIKV RNA was also detected in the semen of mouse models). Importantly, the ratio of cell-free versus cell-associated virus in semen may vary overtime.

Response 2.3:

It is not clear from the comment what particular mechanism that does not rely on production of free viral particles can be responsible for efficient “cell-to-cell infection” and spreading of ZIKV with the semen. To our knowledge, there are no reports that unequivocally demonstrate that sexual transmission of ZIKV can occur through a “cell-associated” virus. Reports of Matusali et al, JCI 2018 and Mahé et al, Lancet Infect Dis 2020 show that several cell types in the semen (i.e., testicular germ cells, exfoliated epithelial cells, and leukocytes) were positive for ZIKV antigens. However, this does not provide an evidence that these cells can support a productive virus replication, secrete the virus into the semen, and contribute to sexual transmission. More importantly, it is not clear whether these ZIKV-positive cells are being infected in a given MRS organ with subsequent shedding into the organ’s luminal compartment, being infected by a cell-free virus while in the liminal/ductal compartments, or if they just simply phagocytosed the virus particles or viral proteins. Such distinctions are important when we analyze the routes of ZIKV spread in the MRS. We believe that the role of such cells in sexual transmission of ZIKV would be purely speculative at the moment, but may warrant future investigations.

- Authors’reply: “Sequestration of cellular microRNAs in the infected cells due to a ‘sponge effect’ can theoretically alter physiological functions of an infected organ. However, ZIKV infection and replication in the target cells inflicts very dramatic changes onto cell physiology (gene expression, membrane architecture and functions, et cetera), eventually leading to cell death due to viral or immune system-mediated factors. In this context, changes in cell physiology inflicted by sequestration of only one type of cellular microRNAs would have a rather negligible effect compared to overall pathogenic changes associated with viral infection”: I disagree with the authors that the changes in cell physiology in their model, whether induced by the miRNA or viral replication, do not deserve attention. This certainly warrants caution as a potential limitation of the approach since any modification of the epididymis epithelial barrier induced by the virus in this immunocompromised model or by the E-cadherin targeting miRNA (E-cadherin playing an important role in the maintenance of the barrier integrity) may directly impact the shedding of viral particles especially in case of polarized shedding (eg allowing the release in the seminal lumen of viruses normally that might be shed only on the basal side of epithelial cells) and hence the results of this study. Although not obvious from the d9 pictures of the organs, if such a dramatic effect of ZIKV infection is observed in this mouse model that any impact of the miRNAs becomes irrelevant even at early time points, this needs to be acknowledged as a limitation to extrapolating the results to other models and to humans in whom such damage is not reported. Importantly also, the potential impact of the miRNAs on the cell physiology deserves to be mentioned in the context of the authors’ vaccines containing miRNA targets.

Response 2.4:

We believe that our sentence “Sequestration of cellular microRNAs in the infected cells due to a ‘sponge effect’ can theoretically alter physiological functions of an infected organ” has been misinterpreted. This may have occurred because we did not do a good job in elaborating this issue. While we accept that the “miRNA sponge effect” may be an important phenomenon when a critical amount of relevant miRNA targets is introduced into the cell, this phenomenon should have a negligible effect (if any) in the context of our in vivo experiments with miRNA targeted viruses. Sequestration of cellular miRNAs through a theoretical “sponge effect” may become physiologically important only when a sufficient concentration of RNA molecules containing relevant miRNA targets is reached in the specific cell. This increase in the number of miRNA targets cannot be happening in our case, since our miRNA targeted viruses were specifically designed to not being able to replicate in the presence of the complementary miRNA (and thus, to produce more copies of miRNA targets). MicroRNA targets inserted in the virus genome are directly responsible for the blocking of virus replication, and therefore, relevant complementary miRNA molecules only can act during the first round of virus replication, when the miRNA targets in viral genome become accessible to them. This will stop virus replication and prevent accumulation of molecules that otherwise could potentially serve as a miRNA “sponge”.

Reviewer #2: “I disagree with the authors that the changes in cell physiology in their model, whether induced by the miRNA or viral replication, do not deserve attention.”

Importance of changes in the cell physiology due to viral replication is a concept that should not be in doubt as being well understood in the field of virology. Regarding “changes in cell physiology ... induced by the miRNA” (we believe that original meaning was “miRNA sequestration and loss of function”) – please see our position described above.

Regarding the differences in the magnitude of ZIKV- induced pathological changes detected in the MRS of AG129 mice and humans, we believe that these differences can be mostly attributed to the immunocompromised status of our mouse model, which has been sufficiently addressed/discussed throughout the manuscript.

- Line 524: miss the word “seminal” before “SIV”

Response 2.5:

Corrected

- Comment on authors’ reply: “ZIKV virions produced by cells of testicular interstitium cannot reach epididymis because they do not have access to the lumen of seminiferous tubules (for that, ZIKV needs to bypass the blood-Sertoli cells barrier and replicate inside the cells of seminiferous tubules)”: of note, the epithelium of the rete testis is considered to be less restricting than that of the Sertoli cells and may enable the leakage of testis interstitial virus and leukocytes into the seminiferous tubules and epididymis.

Response 2.6:

To our knowledge, there are no studies that show that ZIKV can reach luminal side of the testis (or epididymis) via “the leakage” of “interstitial virus”. On the contrary, previous studies by our group (PMID: 30559387, 32614902) and by others (PMID: 29615760, 29378173) demonstrated

a biphasic kinetic of ZIKV replication in the testis, with a second phase being more productive. This strongly suggests that the switch in host cells from the interstitial macrophages (PMID: 29378173) to the cells residing in the seminiferous tubules involves the change of host that supports more productive virus replication. Moreover, we previously showed that ZIKV replication in the cells of the testicular interstitium occurs only during very early phase of testicular infection (~ 3 dpi; see [PMID: 30559387 Fig. 3]), and that all studied viruses (including the scramble positive control and miRNA-targeted viruses) were completely cleared from the testicular (and epididymal) interstitium by 12 dpi (PMID: 30559387, Supplementary Figure 3 and 6), around the time of ZIKV sexual transmission in our experiments. Considering that during late phase of infection there are no (very few) ZIKV- infected cells in the interstitium that can potentially generate virus, it seems unlikely that hypothesized 'leakage' from the interstitium can have a considerable contribution to the total amount of the testis-derived virions/genomes in the epididymis.